# Modeling tissue-scale ciliary transport and mixing in three-dimensional Newtonian flow

Ling Xu[1]*, Pejman Senaei[2], Yi Jiang[2]*

**1** Department of Mathematics and Statistics, North Carolina Agricultural and Technical State University, Greensboro, North Carolina, United States of America, **2** Department of Mathematics and Statistics, Georgia State University, Atlanta, Georgia, United States of America

* lxu@ncat.edu (LX); yjiang12@gsu.edu (YJ)

## Abstract

Mucociliary clearance is the primary defense mechanism in our respiratory system against aerosol pathogens and allergens. The rhythmic movement of cilia on airway-lining cells propels mucus flow, driving the movement of trapped particles. However, the impact of cilia density and distribution on mucociliary mixing and transport at the tissue scale remains poorly understood. In the present work, we present three-dimensional (3D) simulations of ciliary-driven mixing and transport of a Newtonian fluid as an approximation of mucus at the tissue scale. We investigate the influence of ciliary density, cilia cluster spacing, and metachrony on fluid mixing and transport. Our findings reveal that: (i) cilia clusters generate flow swirls whose size scales with ciliary density, (ii) a single cilia cluster generates horizontal and upward transport with horizontal mixing, (iii) optimal spacing between ciliary clusters enhances horizontal transport, and (iv) metachronic waves enhances mixing but reduces net transport. These findings provide useful insight into generic principles of cilia-driven transport in viscous fluids and may inform bio-inspired system design, while further work is needed to extend this work to physiologically realistic mucus transport.

## Author summary

The human airway is exposed to foreign particles, including allergens, pathogens, and aerosolized vaccines or medications. The mucus coating the airway surface traps these particles, and the beating cilia extending from the ciliated epithelial cells play a crucial role in propelling the flow of mucus to clear the trapped particles out of the airway. This mucociliary clearance process is critical for the initiation and progression of airway tissue response, such as allergic responses, infections, and treatment effects. Changes in ciliated cell density or mucus production can significantly impact this process. Conversely, achieving a homogeneous distribution of aerosol drug particles within the airway is essential for optimizing

**Data availability statement:** The source code and data used to produce the results and analyses presented in this manuscript are available from https://doi.org/10.5281/zenodo.17560085 (https://github.com/Jiang-Lab/Mucociliary_Clearance-Tissue_Scale_model).

**Funding:** The author(s) received no specific funding for this work.

**Competing interests:** The authors have declared that no competing interests exist.

treatment outcomes. We simulate cilia-driven mucus flow, using a simplified rod-cilia and Newtonian-fluid model, to examine how the motion of cilia influences the transport and mixing of particles in the mucus. Our key findings suggest an optimal spacing for ciliated cell clusters that enhances directional clearance transport. Intriguingly, while all metachronal waves promotes mixing, its effect on the overall particle clearance depends on the phase lag: some sympletic waves (those travel in the same direction of the forward cilia stroke) enhance transport while majority of the metachronal waves hinder clearance, and can potentially move the particles backwards.

## Introduction

Mucociliary clearance serves as the primary defense mechanism of the airways, playing a crucial role in expelling foreign particles. The airways are lined with a mucus layer that effectively traps inhaled particulates. Beneath the mucus, the cilia, tiny hair-like structures protruding from the airway epithelium, beat rhythmically. This coordinated movement propels mucus, along with trapped particles, out of the airway, contributing to the respiratory defense system [12]. Conversely, when administering nasal sprays for vaccines [27] or treating allergic rhinitis [28], the goal shifts towards achieving maximal mixing with minimal mucus transport.

The airway epithelium primarily consists of the ciliated cells where cilia reside, the goblet cells that are responsible for mucus secretion, and the basal cells, which can proliferate and differentiate to maintain epithelial integrity. Different segments of the airway exhibit distinct cellular compositions [18]. A microscopy image illustrating the distribution of cilia in the rabbit trachea (Sanderson & Sleigh [47]) provides insight into their arrangement. A typical cilia cluster has a dimension of $10\mu m$ and comprises 100-200 cilia on its apical surface [38] (Fig 1). Each cilium is anchored at its base and undergoes periodic beating in a three-dimensional (3D) circular motion, including an effective stroke and a recovery stroke. During the effective stroke, the cilium remains rigid and upright, while it becomes pliable and bends during the recovery stroke [47]. It is well-established that the coordinated beating of cilia generates a net unidirectional flow of mucus, effectively transporting foreign particles or pathogens out of the airway.

Several airway diseases, such as primary ciliary dyskinesia, chronic obstructive pulmonary disease (COPD), and asthma, can impair the ciliary function [33]. Using in vitro human bronchial cultures ranging in size from micrometers to centimeters, Khelloufi et al. found, underneath the mucus, a circular order of the ciliary beating directions (swirl) drives the mucociliary transport and the swirl size was shown to scale with ciliary density [38]. Loiseau et al., also using a reconstituted airway epithelium, showed cilia–mucus hydrodynamic interactions govern the collective dynamics of ciliary-beat directions [39]. Gsell et al. further demonstrated, using human bronchial epithelium reconstituted in-vitro and a 2D hydrodynamic model, that increasing ciliary density increases the swirl size, leading to longer-range unidirectional mucus flow [30]. While these in vitro reconstituted human airway tissues are undergoing

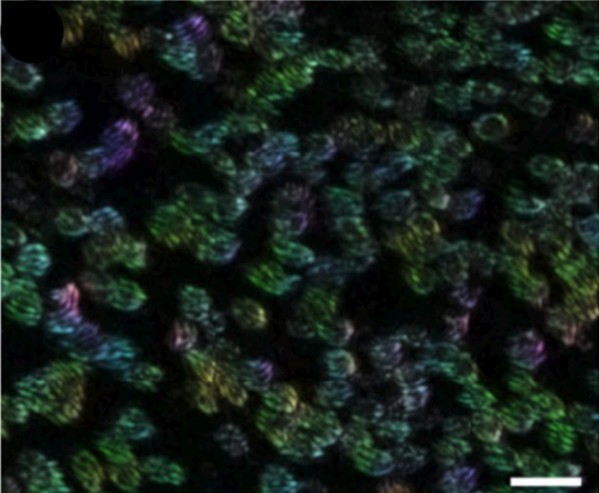

**Fig 1. Tissue-scale cilia image.** Cilia orientation map in human bronchial cultures obtained using over 500 images, from Khelloufi et al. [38]. Scale bar represents $20\mu m$.

ciliogenesis and do not fully represent mature, functional airway tissue, they offer controlled environments to explore how ciliary organization gives rise to mesoscale flow structures, such as mucus swirls.

Many respiratory viruses, including influenza virus and SARS-CoV-2, target the ciliated cells [34,35,37,52,54]. While the lung epithelium can sustain normal function under mild injuries, permanent damage may occur if large areas of ciliated cells die due to infection [18]. SARS-CoV-2 infection, in particular, can inflict severe damage on the lung epithelium, resulting in serious pneumonia, acute respiratory distress, and even death [34,51]. A recent model for within-host SARS-CoV-2 infection incorporates mucociliary clearance as a crucial mechanism in viral transmission [21]. However, how ciliary density changes as infection progresses and how that impacts mucociliary transport has not been fully studied.

The mathematical exploration of the mucociliary system has spanned over six decades since Taylor's work in 1951 [50]. Early investigations focused on the wavy shape of cilium-like organelles, modeling them as sinusoidal functions [29]. Brokaw proposed the shear mechanism involving internal doublet structures to elucidate the undulating motion [15]. Lighthill introduced the slender body theory [36] that represents the beating cilium body as a line of singular forces resulting from the self-activation and self-regulation of the motile body, and the fluid flow velocity near the motile body is proportional to the force exerted. Barton and Raynor made the assumption that feedback from the fluid flow is negligible [10]. They modeled the cilium as a rigid rod, straightening during the effective stroke and automatically shortening during the recovery stroke. Building upon the slender body theory, Cortez introduced the method of regularized Stokeslets, designed to solve the Stokes equations in free space for zero Reynolds number flow [22]. This method has been widely adopted for simulating swimming micro-organelles and other biological fluid flow problems, where inertial forces are negligible [23]. For instance, the Reynolds number within the mammalian respiratory tract is approximately 0.01, where viscous forces play a dominant role in the mucociliary system [55]. Ding et al. [7], using a doubly periodic array of 2D beating cilia, found metachronal wave enhances fluid transport in regions above the cilia tip and mixing in the sub-ciliary region. Guo et al. explored the in-phase and anti-phase synchronization modes of two adjacent cilia [32]. They modeled cilia as chains of beads, each bead is driven by an external periodic force as a rower [53]. Furthermore, Charkrabarti modeled a lattice of cilia as rowers to investigate the formation and maintenance of metachronal waves [17]. Their findings suggest that spatial inhomogeneities contribute to the robustness of the metachronal waves, enhancing long-range transport. Kanale et al. showed metachrony emerges in dense arrays of hydrodynamically interacting cilia and metachronal phase coordination is a stable global attractor in large ciliary carpets [5].

Other mathematical models have delved into various facets of the mucociliary system. Osterman and Vilfan determined optimal beating patterns of cilia based on considerations of energetic efficiency [43]. Studies by Eloy and Lauga have examined the kinematics of the most efficient cilium [25]. Yang, Dillon, and Fauci explored the emergence of cilia beating synchrony through hydrodynamic effects [56], highlighting the complex interactions between fluid dynamics and ciliary activity. We sought to identify key factors influencing cilia motion and particle transport in mucociliary clearance [55]. The advent of supercomputing has enabled large-scale computational studies in this domain. Mitran's work featured a full 3D simulation involving 256 beating cilia [42]. Elgeti and Gompper simulated 400 2D cilia arrays within a 3D fluid flow [24]. Additionally, Ma and Lutchen simulated aerosol deposition using a turbulence model [40]. Their sensitivity analysis highlighted the crucial dependence of simulation outcomes on the characteristics of the flow field [40].

In all these mathematical models, mucus has been treated as an incompressible viscous fluid. However, airway mucus is a sticky gel composed of mucin and periciliary-glycocalyceal layer [4]. While these studies collectively underscore the importance of computational models in unraveling the nuances of cilia-driven flow, it is important to remember the difference to physiologically accurate mucociliary system in airway.

The objective of this study is to address how ciliary density and distribution impact the mucociliary transport and mixing at the tissue level. We adopt the regularized Stokeslet method and assume an incompressible Newtonian fluid to approximate mucus flow, following standard modeling practices [45]. We acknowledge that mucus is a complex viscoelastic gel and that Stokes flow models do not capture its polymeric structure or time-dependent rheology. However, this assumption allows us to isolate and analyze key geometric and kinematic effects of ciliary distribution and coordination.

Extending from our previous work [55], we model the 3D cilium beating as a rigid rod with a prescribed motion. We vary the cilia cluster spacing, density, and metachronality, to study their impact on the trajectories of massless particles placed in the flow. These particles not only help to illustrate material transport and mixing, but also mimic the motions of tiny particles such as viruses whose sizes are negligible. Our simulations unveil intriguing global transport phenomena and diverse local mixing patterns. Additionally, we examine the phase shifts of metachronal waves within cilia clusters, discovering that such shifts can hinder overall transport.

## Methods

To simulate mucociliary flow in 3D at the tissue scale, we integrate a cilium beating model and a mucus flow model. As we do not consider the multilayer composition of mucus and periciliary liquid, nor the complex composition of airway epithelium, our model is closer to the artificial biomimetic cilia constructed in the lab [8]. We set up the computation model of cilia clusters, and vary ciliary density and distribution.

### Cilium beating model

We model the cilium as a rigid rod with a length $L$ that changes during the forward and backward strokes in cilium beating motion [55]. Fig 2a illustrates the motion of the rod in one revolution. The position of rod $\boldsymbol{x} = (x, y, z)$ is

$$x(t) = x_0 + s(t) \cos \theta(t) \cos \phi(t), \tag{1a}$$
$$y(t) = y_0 + s(t) \cos \theta(t) \sin \phi(t), \tag{1b}$$
$$z(t) = z_0 + s(t) \sin \theta(t), \tag{1c}$$

$\boldsymbol{x}_0 = (x_0, y_0, z_0)$ is the base of the rod on the $(x,y)$-plane ($z_0 = 0$), $s \in [0, L]$ is the Lagrangian parameter. The instantaneous velocity of the rotating rod is computed analytically using $\boldsymbol{u} = \mathrm{d}\boldsymbol{x}/\mathrm{d}t$.

Figs 2b and c show the rod's profile projected onto the $(x, z)$- and $(x, y)$- planes. The rod rotates at a given angular velocity $\omega$ in the counter clockwise direction about $z$-axis. The azimuthal angle $\phi$ and the polar angle $\theta$ both time-dependent. The azimuthal angle is $\phi(t) = \phi_0 + \omega t$, $\phi_0$ is the initial phage lag of the rod at time $t = 0$. The polar angle is $\theta(t) =$

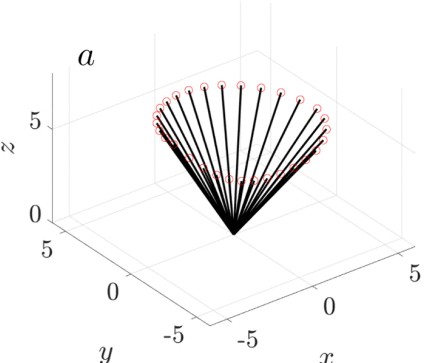 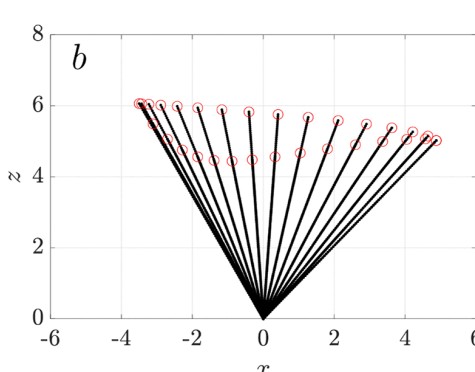 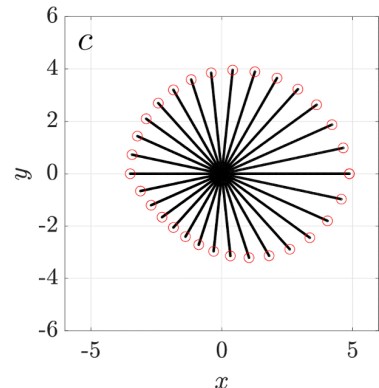

**Fig 2. Illustration of a 3D beating cilium simplified as a rotating rod with varying length.** (a) The rod's motion over one full revolution, with red circles marking the rod tips. (b) Projection of the rod's profile onto the $(x,z)$-plane, (c) Projection of the rod's profile onto the $(x,y)$-plane.

$\theta(\phi(t))$, and it describes the angle from the $(x,y)$-plane. The range of the polar angle $\theta$ is limited to $[0.8, 2.4]$ radians based on the experimental depiction of a typical beating cycle of a cilium [47].

We previously simplified the cilium as a rigid rod with a varying length $L(\phi)$ to mimic the forward (power) stroke and the backward (recovery) stroke [55]. We adopt the same description here:

$$L(\phi) = \begin{cases} 7, & \phi \in [0, \pi] \text{ forward,} \\ -0.3235\phi^3 + 5.2009\phi^2 - 26.6650\phi + 49.4710, & \phi \in [\pi, 2\pi) \text{ backward,} \end{cases} \tag{2}$$

where $L$ is longer during the forward stroke than the backward stroke. With counter-clock-wise rotation, the effective stroke is in the negative $x$ direction, which is the preferred direction of clearance. As shown in Fig 2c, the trajectory of the rod tip is not circular on the $(x, y)$-plane due to the variation of both $\theta$ and the rod length.

### Cilia-driven mucus flow

The mucus is modeled as an incompressible Newtonian fluid, where the inertial forces are negligible [57]. The governing equation is:

$$-\nabla P + \mu \nabla^2 \boldsymbol{u} + \boldsymbol{F} = \boldsymbol{0}, \quad \nabla \cdot \boldsymbol{u} = 0, \tag{3}$$

where $P$ is the fluid pressure, $\boldsymbol{u}$ is the fluid velocity, $\mu$ is the fluid viscosity, and $\boldsymbol{F}(\boldsymbol{x})$ represents the force density exerted on the fluid by the cilia. Following the work of Guo et al. [32], we choose a regularized Stokeslet formulation [45] where forces are distributed along the rod. The force density at a point $\boldsymbol{x}_c$ follows a distribution

$$\boldsymbol{F}(\boldsymbol{x}) = \boldsymbol{f}\psi_\epsilon\left(|\boldsymbol{x} - \boldsymbol{x}_c|\right), \quad \text{where} \quad \psi_\epsilon(r) = \frac{15\epsilon^4}{8\pi(r^2 + \epsilon^2)^{7/2}}.$$

Here $\boldsymbol{f}$ is the force coefficient, $\psi_\epsilon(r)$ is the regularized kernel, and $r = |\boldsymbol{x} - \boldsymbol{x}_0|$. We choose $\epsilon = 0.1$, which corresponds to the cilium thickness. The fluid flow is subject to no-slip condition at the bottom wall or epithelial cell surface, $\boldsymbol{u} = \boldsymbol{0}$ at $z = 0$. As the stiff rod beats, the force coefficients $\boldsymbol{f}$ of all Stokeslets are updated according to the instantaneous rod velocity.

## Tissue-scale cilia clusters

To simulate multiscale spatial heterogeneity of cilia organization at the tissue scale, as observed in airway epithelia [3], we begin with a single cilia cluster measuring 8 $\mu$m $\times$ 8 $\mu$m. Within each cluster, 3 $\times$ 3 model cilia [8] are arranged in a regular grid with uniform intra-cluster spacing of $D = 4$ $\mu$m. We then construct a linear array of three such clusters, varying the inter-cluster spacing $D_c$ to study the effects of spatial separation (Fig 3a). This array is aligned along the *x*-axis, which represents the longitudinal axis of the airway; particle transport along this direction – specifically in the negative *x* direction, corresponding to mucociliary clearance – is the primary focus of our analysis. Because a single ciliated cell in the human airway epithelium typically hosts approximately 100–200 motile cilia [49], each model cilium in our simulations can be interpreted as representing the collective, synchronized beating of multiple cilia on a single ciliated cell.

Fig 3b illustrates a larger area of the epithelium containing multiple ciliated clusters. The area is represented as a square with side lengths of $L_x = L_y = 100\mu$m. The ciliary density, $\nu$, is the ratio of the total area occupied by the cilia clusters (small squares) to area of the larger square region. These cilia clusters are distributed randomly within this region. The locations of ciliary clusters are randomly sampled from a uniform distribution. For each cilia density, three independent realizations of cilia cluster placements are used. In simulations with metachronal waves, each column of three cilia in the cluster are asynchronized, and a constant phase-lag $\phi_0$ is imposed.

Previous studies have shown that spatial disorder in ciliary patch arrangement can enhance mucus clearance due to the emergence of large-scale coordinated flow [3,5]. Our model builds on this insight to investigate how cilia density and metachronal coordination impact fluid transport and mixing in a disordered ciliary carpet.

Table 1 lists all computational parameters used in this study.

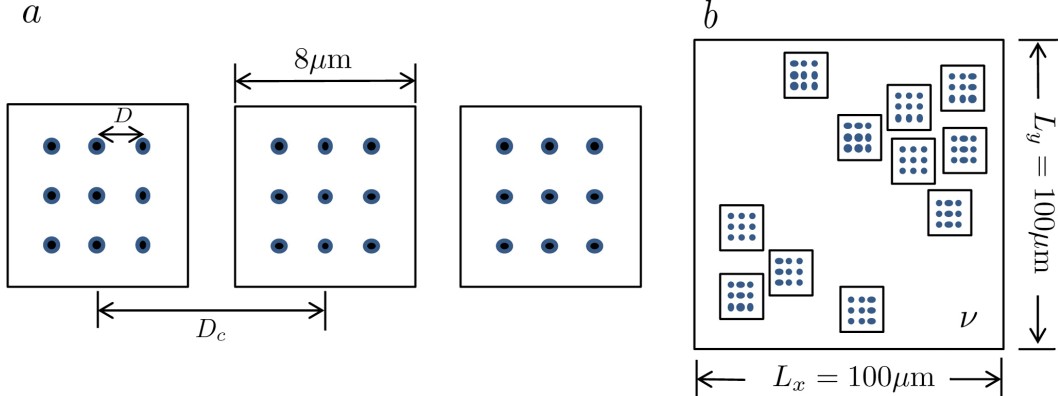

**Fig 3**. **Illustration of the tissue-scale cilia cluster model.** (a) The top view of an array of three cilia clusters, each containing nine equally spaced cilia. The spacing between individual cilia within a cluster is $D = 4\mu$m, while the inter-cluster spacing is $D_c$ (variable). (b) A large patch containing multiple clusters, covering an area of $100\mu$m $\times$ $100\mu$m. $\nu$ is the variable ciliary density. Blue dots indicate the position of individual cilia.

**Table 1**. **Parameters used in this study are based on experimental measurements.**

| Parameter | Symbol | Dimensional value | Reference |
|---|---|---|---|
| Maximum rod length | max($L$) | $7\mu$m | [49] |
| Rod/Cilium beating frequency | | 18 Hz | [49] |
| Rod/Cilium angular velocity | $\omega$ | $36\pi s^{-1}$ | |
| Rod spacing in the cilia cluster | $D$ | $4\mu$m | |
| cilia cluster spacing | $D_c$ | $3D$-$12D$ | |
| Ciliary density | $\nu$ | 0.1,0.2,0,4 | [38] |
| Tissue patch | $L_x, L_y$ | $100\mu$m | |

## Results

### In synchrony, transport and mixing scale with ciliary density

To investigate particle motion driven by mucociliary flow at the tissue scale, we simulate a patch of ciliated cells represented by a square region spanning $[-50, 50] \times [-50, 50]\mu$m. Previous models of ciliary carpet have considered a uniform ciliar array (e.g., [7]). Our model explicitly considers the random x-y placement of cilia clusters to mimic the experimentally observed heterogeneity in ciliated cell distribution in mouse trachea [3] and reconstituted human airway tissue cultures [30,38,39].

The coordinated beating of these cilia drives fluid flow. At a low cilia density ($\nu = 0.1$), the 3D velocity field of the cilia patch shows local rotational order (Fig 4), resulting in less predictable transport directions.

To examine how the swirl motion drives particle distribution and its dependence on ciliary density, we compare mucociliary flow patterns at three ciliary densities: $\nu = 0.1, 0.2$, and $0.4$. The ciliated cell density in human bronchial culture ranges from 0 to 0.70, depending on the location in the airway and pathological conditions, with an overall average of 0.15 [38]. The ciliated cell density in mouse airway averages around 0.37 with a large spatial heterogeneity [3]. Our density choices reflect this physiological range.

Fig 5 presents top-down views of passive particles in a ciliated tissue patch, showing the time evolution of particle trajectories for three different ciliary densities, $\nu = 0.1, 0.2$, and $0.4$ (rows a–d, e–h, i–l, respectively). Each simulation features synchronously beating cilia clusters, with their spatial locations shown in the top row (a, e, i). The initial particle distribution is color-coded into four groups (black, blue, green, red) within a central square domain $[-50, 50] \times [-50, 50]\mu$m and confined vertically to $z \in [5, 9]\mu$m. Over time, particles are advected by the cilia-driven flow, particle distributions evolve from ordered arrays to dynamically stretched and rotationally mixed configurations. Depending on ciliary density, particles exhibit different transport patterns. At low density ($\nu = 0.1$), particles remain relatively localized, with weak flow-induced deformation and limited mixing by $t = 11.11$ s (400 cycles). At intermediate density ($\nu = 0.2$), particle trajectories begin to form interconnected swirls, with moderate spreading and intermixing. At the highest density ($\nu = 0.4$), the flow becomes more coherent and global, giving rise to large-scale rotational patterns that facilitate pronounced transport and mixing across the domain. Notably, the number of swirls is substantially fewer than the number of clusters, suggesting that the rotational structures emerge from collective hydrodynamic interactions rather than from isolated cluster activity.

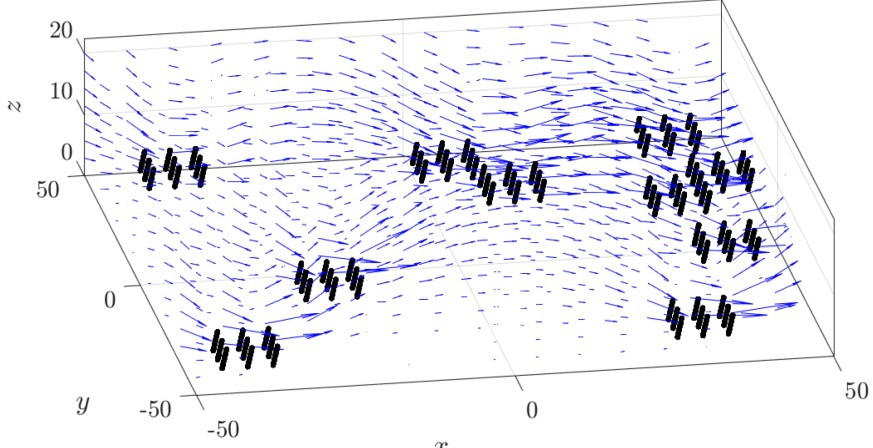

**Fig 4**. **Typical snapshot of the swirl pattern in viscous flow driven by cilia at a density of** $\nu = 0.1$. Blue arrows indicate the velocity field. Black lines represent cilia, beating synchronously with an initial phage $\phi_0 = 0$.

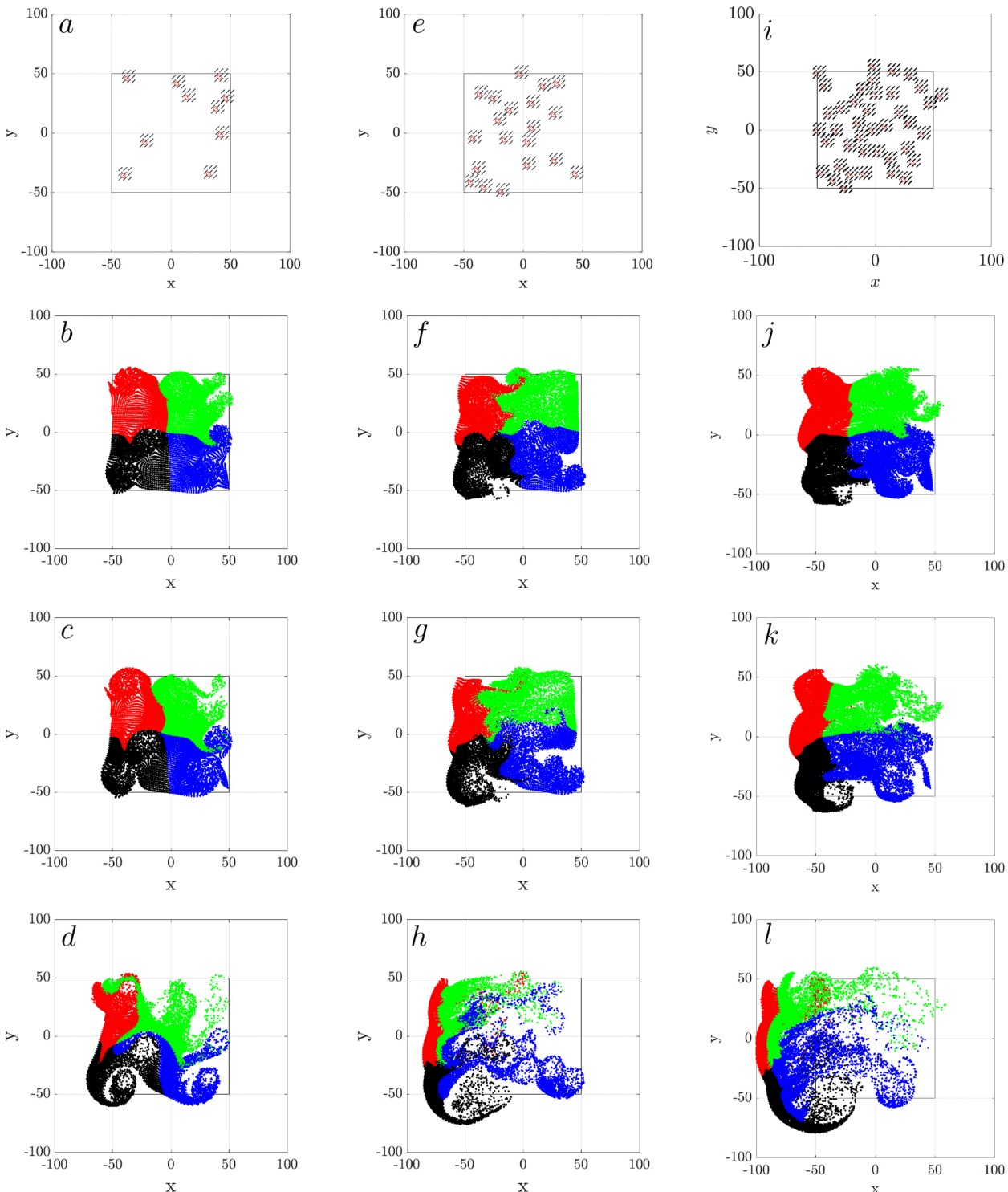

**Fig 5. Swirl motion of passive particles as a function of ciliary density: (a-d)** $\nu = 0.1$ **(e-h)** $\nu = 0.2$, **(i-l)** $\nu = 0.4$. The top row (a, e, i) shows the locations of cilia clusters. The remaining rows show the top view of the distribution of passive particles at $t = 1.38s$ (b, f, j), $t = 2.78s$ (c,g,k), $t = 11.11$ s (d, h, l). These time points correspond to 50, 100, and 400 cilia beating revolutions, respectively, with all cilia beating synchronously. Particles are initially distributed in four color-coded groups (black, blue, green, and red) within the ciliated region $[-50, 50] \times [-50, 50]\mu$m and $z \in [5, 9]\mu$m. The simulation domain is $[-100, 100] \times [-100, 100]\mu$m. Each time sequence shows a representative simulation out of three independent, random realizations.

PLOS Computational Biology

We calculate the center of mass (CoM) of all particles to quantitatively compare transport efficiency as a function of cilary density (Fig 6). The results for each density value ($\nu$) represent the average of three independent simulations. For all three densities, the net particle transport occurs in the negative *x*- and *y*-directions and the positive *z*- direction, indicating particles movement toward the lower-left corner of the simulation domain and away from the cell surface. Higher ciliary density results in faster and more extensive directed transport, consistent with in vitro observation [30].

To quantify the spatial organization of passive particles seen in the bottom row of Fig 5, we calculate Ripley's K function $K(r)$ for massless particles near the cilia tip at $z = 7$ $\mu m$, where rotational flow is strongest (Fig 7a). Ripley's K function provides a scale-dependent measure of spatial clustering: values above the reference line for complete spatial randomness (CSR) indicates aggregation, while values below indicate dispersion [20]. We see that for all ciliary densities, $K(r)$ initially exceeds the CSR reference line, indicating short-range clustering due to swirls. The curves cross the CSR line at a critical radius $r_0$, which measures the cluster size, beyond which particles become more uniform or dispersed. We measure $r_0 \approx 20$ for $\nu = 0.1$, $r_0 \approx 28$ for $\nu = 0.2$, and $r_0 \approx 32$ for $\nu = 0.4$. These clustering radii correspond to the particle swirl sizes in the final frames of Fig 5, supporting the interpretation that particles become temporarily trapped with swirls before dispersing. When all the particles in 3D domain are considered, the $K(r)$ values fall consistently below the CSR line for all densities (Fig 7b), indicating global dispersion at late times. This finding highlights that local vortex structures (swirls) induce

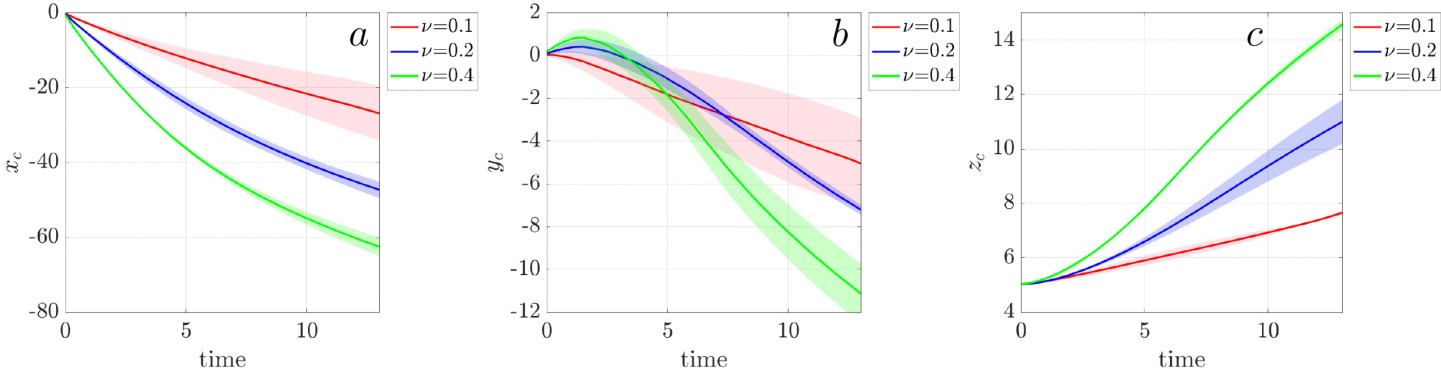

**Fig 6**. **Net particle transport driven by mucociliary flow at ciliary densities $\nu = 0.1$ (red), $\nu = 0.2$ (blue), and $\nu = 0.4$ (green).** (a) Center of mass (CoM) displacement in the *x*-direction, (b) CoM displacement in the *y*-direction, (c) CoM displacement in the *z*-direction. For each density $\nu$, three independent simulation were performed. The solid lines represent the mean displacement and the shaded regions are the standard deviation.

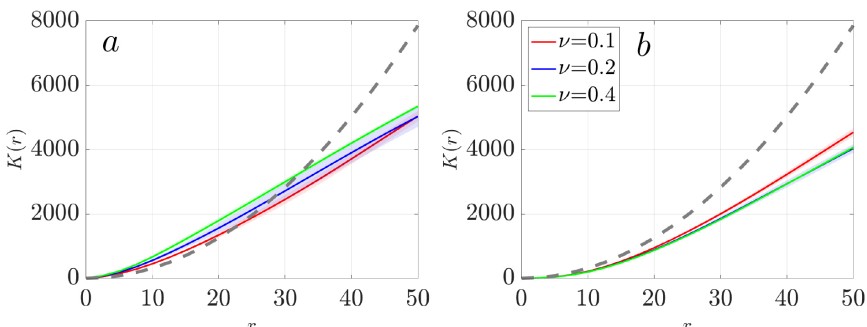

**Fig 7**. **Ripley's K function for particles at three ciliary densities $\nu = 0.1$ (red), $\nu = 0.2$ (blue), and $\nu = 0.4$ (green) at $t = 11.11$ s.** (a) Particles initially located at a height of $z = 7\mu$m. (b) All particles within the vertical range of $z = [0, 10]\mu$m. The gray dashed reference line indicates complete spatial randomness. For each density $\nu$, three independent simulation were performed. The solid lines represent the mean displacement and the shaded regions are the standard deviation.

mesoscale clustering, the overall dynamics eventually promote large-scale mixing and dispersal, particularly at high cilia densities.

In the human airway, effective mucociliary clearance relies on preventing pathogenic particles from reaching the epithelial surface by keeping them trapped in the mucus layers. In contrast, for aerosolized drug delivery, the desirable outcome is the opposite: rapid transport of agents toward the cells. Therefore, understanding how cilia motion drives vertical movement is important to both clearance and delivery.

Fig 8 illustrates vertical mixing at a ciliary density of $\nu = 0.1$. Massless tracer particles are initially partitioned into two groups: the upper half (cyan) and the lower half (black). The particle positions are shown at $0, 50, 100, 200,$ and $400$ cilia beating cycles in the $x$-$z$ plane. As time progresses, vertical mixing is evident: the upper (cyan) particles migrate downward, while the lower (black) particles move upward. Concurrently, the particle distribution expands vertically, growing from an initial height of $10\mu$m to about $20\mu$m.

To determine the effect of ciliary density on vertical mixing, we compare the vertical particle distribution for three densities $\nu = 0.1, 0.2,$ and $0.4$, at $400$ cilia beating cycles (Fig 9). We use the mixing number $m$ to quantify mixing between initially segregated groups, as previously used by Stone & Stone [6] and Ding et al. [7]:

$$m = \left( \prod_{i=1}^{N_1} \min_{1 \leq j \leq N_2} |\mathbf{z}_i - \mathbf{z}_j|^2 \right)^{1/N_1} , \tag{4}$$

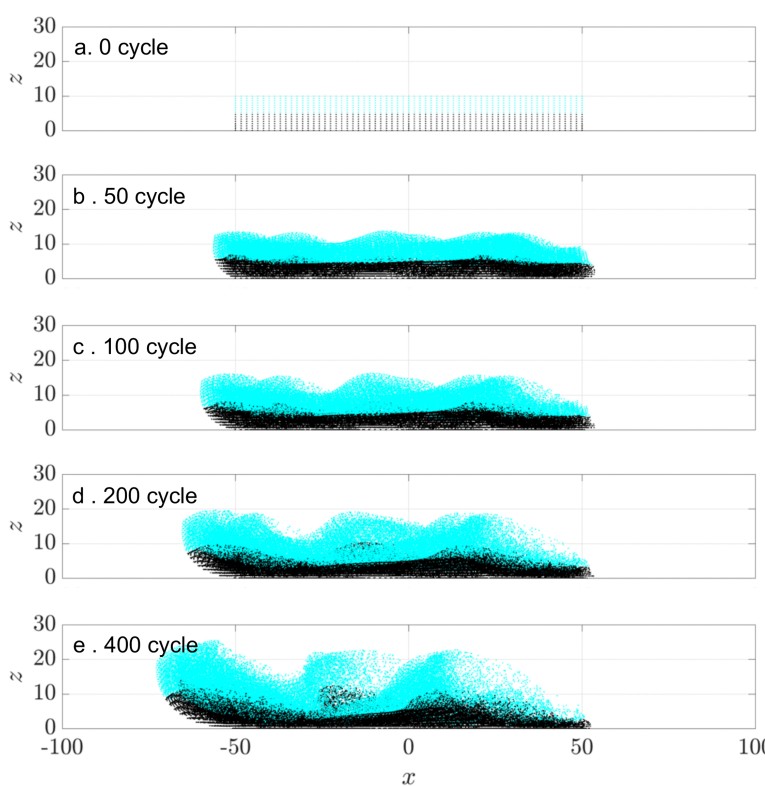

**Fig 8**. **Vertical mixing at tissue level with ciliary density** $\nu = 0.1$. Particles are initially partitioned into two groups: the bottom half (black) and the top half (cyan), to visualize vertical stratification and subsequent mixing. (a–e) Snapshots of particle distributions at $t = 0, 50, 100, 200,$ and $400$ cilia beating cycles, shown for a phase lag of $\phi_0 = 0$. All cilia beat synchronously.

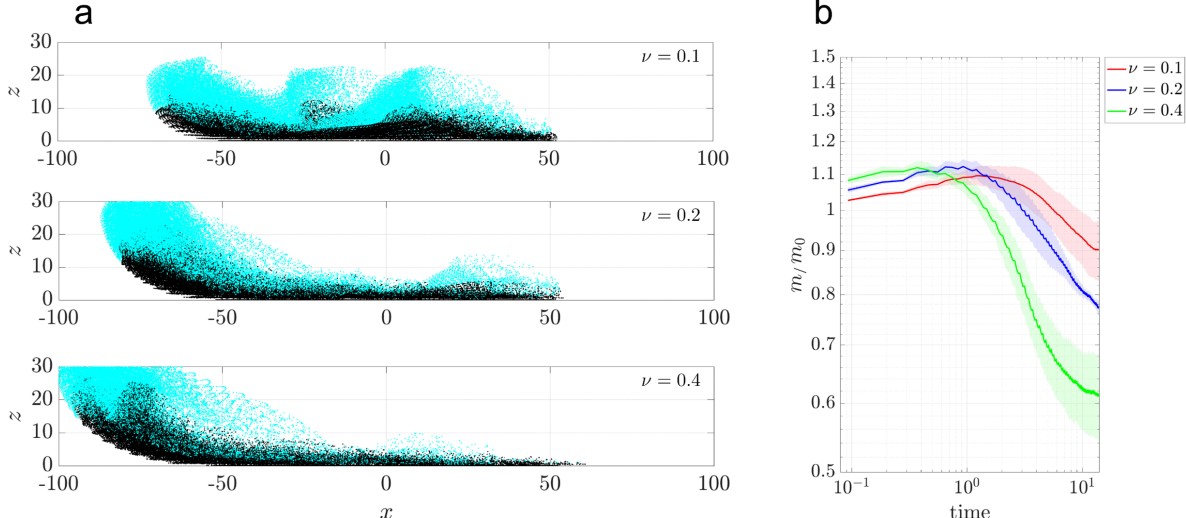

**Fig 9**. **Vertical mixing at tissue level with ciliary densities** $\nu = 0.1, 0.2, 0.4$. Particles are initially partitioned into two groups: the bottom half (black) and the top half (cyan). (a) Snapshots of particle distributions after 400 cilia beating cycles ($t = 11.11$ s, for $\nu = 0.1, 0.2, 0.4$. (b) Normalized mixing number $m/m_0$ vs. time on a log-log scale. For each density $\nu$, three independent simulation were performed. Solid lines represent the mean values and shaded regions represent the standard deviation. All cilia beat synchronously ($\phi_0 = 0$).

where $N_1$ and $N_2$ represent the number of particles in each group, $\mathbf{z}_i$ is the particle's coordinate. The mixing number $m$ is large for complete segregation and decreases in value for increased mixing. Fig 9a shows that particles are transported farther in the negative $x$-direction at larger ciliary densities, as expected. Vertical mixing is evident at all three densities, with upper-layer particles migrating toward the bottom surface, and vice versa. The mixing numbers $m$ decreases over time for all ciliary densities (Fig 9b), indicating progressive mixing. At short time ($t \leq 1$), $m$ initially increases due to the particle scattering in the open half-plane computational domain. As ciliary density increases, the greater number of beating cilia injects more energy into the system, accelerating the mixing process. Three independent random realizations of ciliary clusters confirm the same overall trend.

From a hydrodynamic standpoint, the ciliated surface acts as a distributed vorticity source. Each cilium or ciliary cluster generates localized shear near the epithelial surface, injecting vorticity into the surrounding fluid. To analyze the hydrodynamic interactions between local vortices, we highlight a few particle trajectories (Figs 10a-c), a small sample from those shown in Fig 5. In addition, we measure the time-averaged velocity field over a $2\pi$-beating cycle (Figs 10d-f), and their vorticity field on the $xy$-plane (Figs 10g-i), for three cilia patch densities, $\nu = 0.1, 0.2,$ and $0.4$.

For all densities, we see over time, particle groups are increasingly stretched, rotated, and mixed, and the net particle transport results from the combined contributions of all discrete cilia clusters. Increasing ciliary density ($\nu$) alters the flow structure and particle transport dynamics.

At low density ($\nu = 0.1$), trajectories remain mostly localized and disordered, with limited particle transport or mixing beyond the immediate vicinity of the clusters. As density increases to $\nu = 0.2$, the paths begin to exhibit partial coherence and mild circulation. At $\nu = 0.4$, the trajectories form large, coherent swirl patterns, indicating strong collective hydrodynamic interactions between cilia clusters.

These visual trends in particle motion are supported by the velocity fields shown in the middle row (d–f). At low density, the velocity field is fragmented and spatially confined. Intermediate density shows moderate alignment and flow extension, while high density yields a clear, directed flow pattern across the domain—resembling a bulk mucus transport.

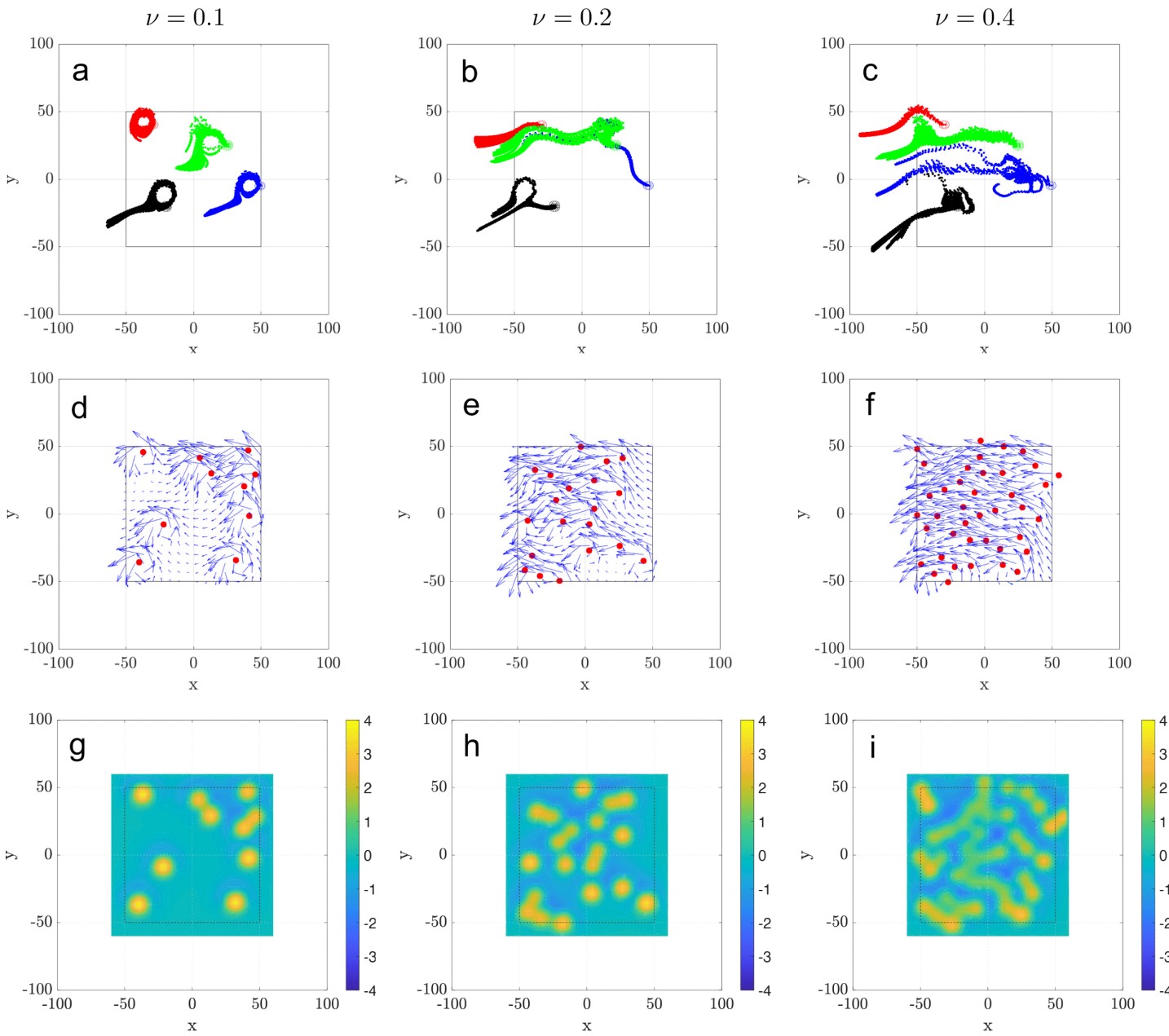

**Fig 10. Particle trajectories, velocity field, and vorticity distributions at tissue level: (a-c) Trajectories for selected particle over 400 ciliary beating cycles for three densities** $\nu = 0.1, 0.2, 0.4$. Initial locations indicated by empty circles. (d-f) Time-averaged (over one cilia beating cycle) velocity vector fields in the $xy$-plane near the cilia tips ($z = 7$ $\mu$m). (g-i) Corresponding vorticity maps, on the $x$-$y$ plane, showing the spatial distribution of rotational flow. All cilia beat synchronously with phase lag $\phi_0 = 0$.

The bottom row (g–i) further confirms this interpretation with the corresponding vorticity fields. At $\nu = 0.1$, vorticity is concentrated in isolated patches. As density increases, neighboring vorticity regions merge, giving rise to extended, mesoscale rotational structures. This transition signifies the emergence of swirls—vortex-like flow domains that organize local particle motion into globally coordinated patterns.

These panels together demonstrate that increasing ciliary density not only enhances overall transport but also induces qualitatively distinct flow regimes: from disordered and localized at low density to coherent and swirl-dominated at high density. These findings are consistent with experimental observations in human bronchial cultures and support the hypothesis that swirl formation is an emergent consequence of hydrodynamic coupling among densely packed cilia clusters.

Taken together, we have a clear qualitative understanding on how increasing ciliary density ($\nu$) alters the flow structure and particle transport and mixing dynamics in a tissue-scale patch of ciliated epithelium. At low density, the lack of collective flow limits the spatial scale of swirls. Flow is too weak and uncoordinated. As ciliary density increases, hydrodynamic interactions between neighboring clusters generate larger, more coherent vortices, which merge into mesoscale swirls (larger than the individual clusters), leading to directed-transport and mixing. This is consistent with previous experimental observations [3,30,38,39], where higher cilia density correlates with the emergence of large-scale mucus swirls.

### Transport and dispersion in a single cilia cluster

Next, we examine in detail how increasing ciliary density enhances both transport and mixing. To begin, we analyze the particle movement with a single cluster of $3 \times 3$ evenly spaced cilia, as illustrated in Fig 3a.

Fig 11 shows the velocity field generated by a single cluster of cilia over one beating cycle, in both top (a-d) and side (e-h) views. Viewed from the top (a-d), the velocity field has a clear four-lobed symmetry, with outward flows directed along the cardinal axes: upwards, leftwards, downwards, and rightwards, reflecting the coordinated sweeping motion of the cilia. The induced flow extends beyond the cluster boundaries and gradually decays radially outward. In the side view (Figs 11e-h), the velocity field from the base to the tip of the cilia provides a more dynamic perspective. Near the base of the cilia (cell surface), velocities approach zero due to the no-slip boundary. The magnitude of velocity increases with height and peaks between $z = 5 \, \mu$m and $10 \, \mu$m, near the cilia tip during the forward stroke. Interestingly, at $\phi = \pi$ (Fig 11g), when the cilium initiates the recovery stroke and begins to retract, the velocity field appears less structured in the side view despite retaining a symmetric pattern from above (Fig 11c). This apparent discrepancy arises because the velocity vectors at this phase largely direct out of the plane of the side view, perpendicular to the page, reducing their in-plane projection. This phase coincides with the shortening of the cilium during the backward stroke, as described by the time-dependent geometry in Equation (2).

Fig 12 shows the distribution of passive particles from both top and side views at four time points: $t = 0$ s, 1.38 s, 2.78 s, and 11.11 s, corresponding to 0, 50, 100, and 400 cilia revolutions, respectively. The particles are color-coded by height ($z$) as blue, red, and black. From the top view, a single counterclockwise swirl emerges as early as $t = 1.38$ s, after only 50 cilia revolutions. The swirl grows in time, extending beyond the cilia cluster boundary. As expected, the bottom (black) particles exhibit minimal motion, the mid-height (red) particles move less than the top (blue) particles. Overall, we see a net transport of particles to the lower left, in the negative $x$ and $y$ directions.

From the side view, the vertical mixing of the blue and red particle layers is evident. The mixing pattern is non-uniform, as indicated in the black box around $x \in [-10, 2] \, \mu$m in Fig 12h, where the blue and red particles intermingle more extensively than in regions outside the black box. Despite this non-uniformity, all particles exhibit an upward movement trend in the $z$ direction.

We further calculate the Ripley's K function to quantify the spatial distribution of the particles at $t = 11.11$ s. Fig 13 shows $K(r)$ for particles at different heights $z(\mu$m). Particles at $z = 1$ exhibit minimal movement, resulting in a curve (black) that falls below the reference CSR line. Particles at all heights $z > 1(\mu$m) display clustering, as their K function curves lie above the reference line for small r values. The curves cross the reference CSR line at critical radii $r_0$. For $r > r_0$, particles show dispersion as their curves fall below the reference line. This pattern of short range clustering and longer range dispersion is associated with the particle swirls observed in Fig 12. Clustering is most pronounced at $z = 5$ and $z = 7$, near the cilia tips.

PLOS Computational Biology

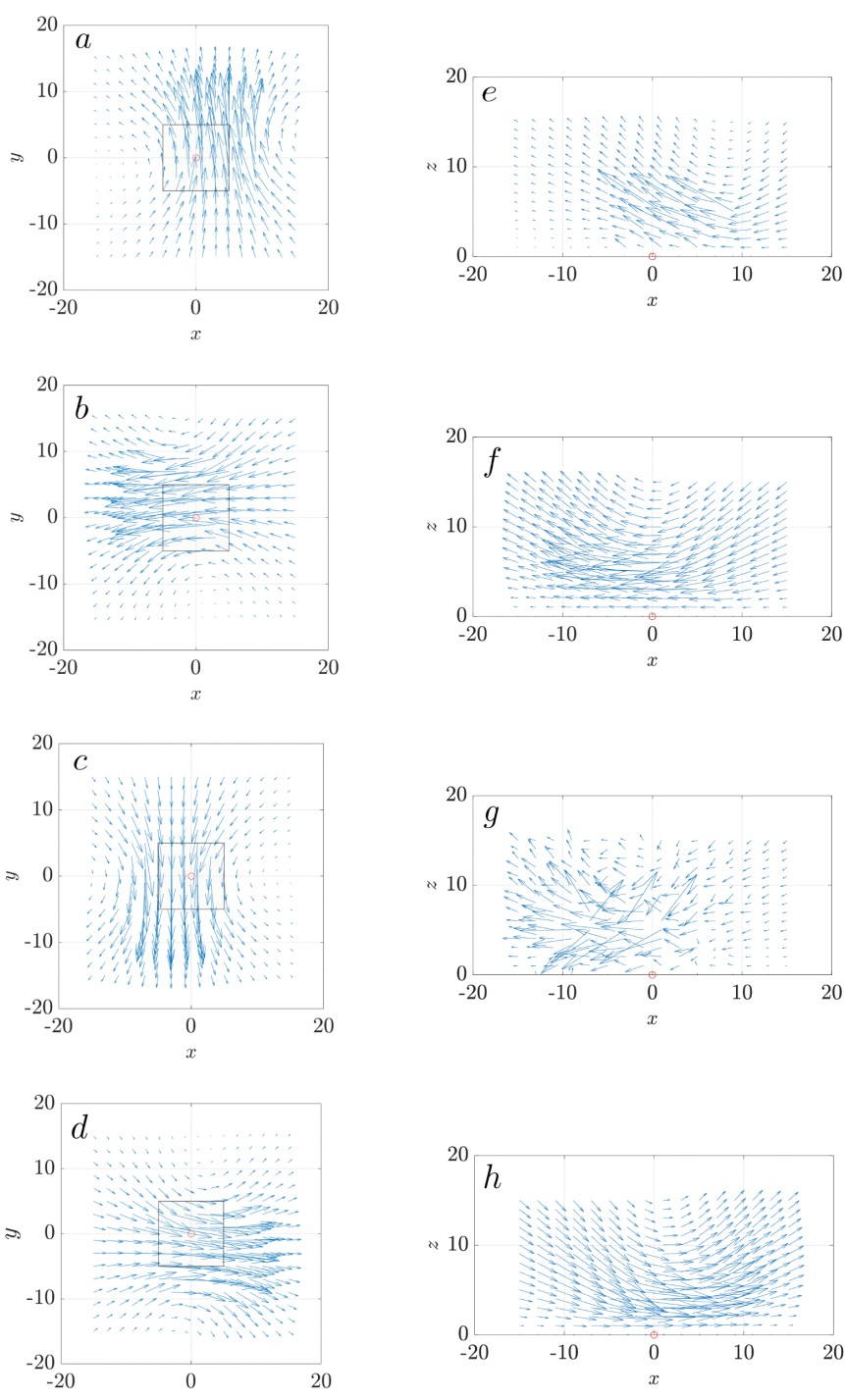

**Fig 11. Velocity field generated by a single cilia cluster over one full beating cycle.** (a-d) Top-down views at $z = 5$ $\mu$m at four phases of the cilia beating cycle ($\phi = 0, \pi/2, \pi, 3\pi/2$). (e-h) Corresponding side views at $y = 0$ $\mu$m. The black box marks the boundary of the cilia cluster, with the cluster center located at (0,0) (red circle). All cilia beat synchronously with an initial phase angle of $\phi_0 = 0$.

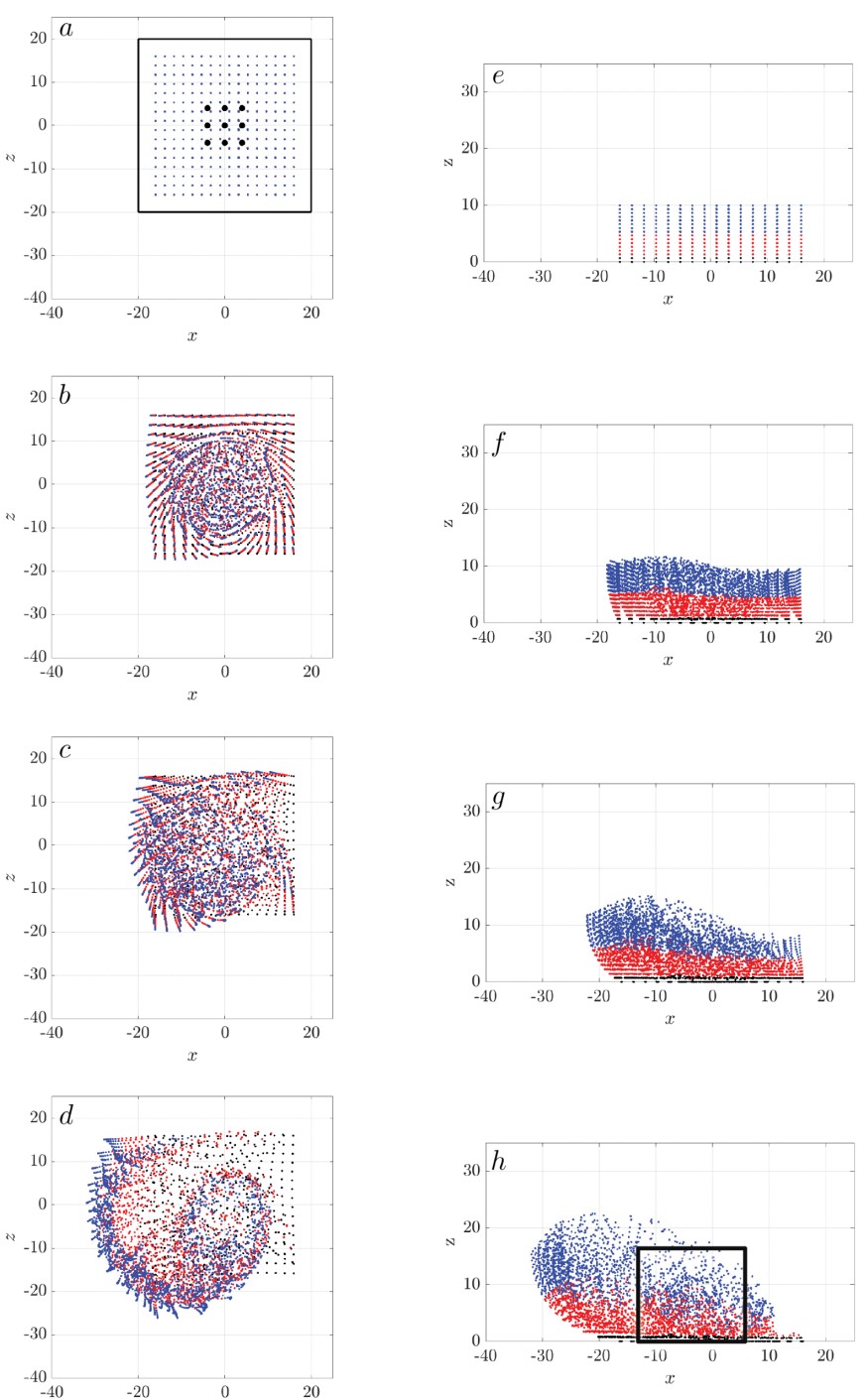

**Fig 12. Particle swirl motion generated by a single cilia cluster: Distribution of passive particles visualized in top view (left column: (a-d)) and side view (right column: (e-h)).** Time progresses from top to bottom, with snapshots at $t = 0$s in (a) and (e); $t = 1.38$s in (b) and (f); $t = 2.78$s in (c) and (g); and $t = 11.11$ s in (d) and (h). These time points correspond to 0, 50, 100, and 400 cilia revolutions, respectively. The black box in (a) outlines the boundary of the cilia cluster boundary, and the blue dots indicate the positions of the 9 individual cilia. In (e), particles are color-coded by their height ($z$): blue, red, and black represent different elevation range. All cilia beat synchronously with initial $\phi_0 = 0$.

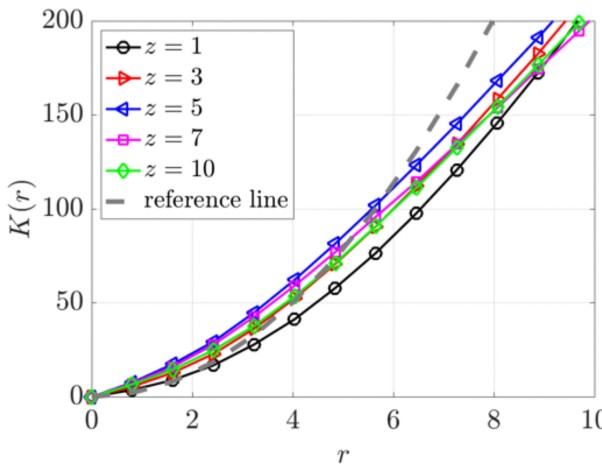

**Fig 13. Ripley's K function for particles at various heights ($z = 1, 3, 5, 7, 10$)$\mu$m at $t = 11.11$ s for a single cilia cluster.** The gray dashed reference line indicates complete spatial randomness.

## Maximal transport in array of synchronized cilia clusters with optimal spacing

As each cilia cluster generates a swirl that shows a net transport and mixing, how do cilia clusters interact with each other? Next, we examine the transport and mixing with an array of three synchronized cilia clusters. We vary the spacing between clusters, $D_c$, from $3D$ to $12D$, with $D = 4\mu$m, as illustrated in Fig 3. When $D_c = 3D$, these three cluster boxes are adjacent to each other, forming one continuous cluster three times as long. Consequently, this arrangement establishes an extended platform for the velocity field, leading to a more pronounced and uniform motion in the $x$ direction (Fig 14).

We color the passive, massless tracer particles based on their initial $x$ positions to visualize their motion induced by three adjacent cilia clusters. The top-view plots of the particles (Figs 15a-d) show the formation of a counterclockwise swirl, similar to the pattern observed with a single cilia cluster (Figs 12a-d) but on a larger scale. Over time, the swirl expands and travels toward the lower-left direction, accompanied by significant horizontal mixing as particles of different colors intermix.

In contrast, the side-view plots (Figs 15e-h) show a predominant movement of green particles, initially positioned at the far right, migrating towards the left. Remarkably, particles from all four color groups exhibit significant mixing in the $x$ direction, as evidenced by the presence of all colors near the left boundary (Fig 15h). This pronounced mixing and transport result from the close proximity of the three cilia clusters, which collectively behave like a single large cluster, facilitating efficient mixing and transport.

At the opposite end of the spacing spectrum, with the cluster spacing set to $D_c = 12D$, the clusters are sufficiently separated to each generate individual swirls (Fig 16). Over time, the swirls expand into adjacent areas, initiating particle mixing at the interfaces. The particles move upwards. But unlike the closely spaced clusters previous described, the majority of the four particle colors remain largely distinct along the $x$ direction, indicating limited mixing between them.

It is now evident that cluster spacing significantly influences particle transport dynamics. We see limited net particle transport at both extreme cluster spacings of $D_c = 3D$ and $12D$. To determine if there is an optimal cluster spacing that maximizes net transport, we vary $D_c$ between $3D$ and $12D$ and analyze the net transport as measured by the center of mass of all the particles. As shown in Fig 17a, the maximal transport in $x$-direction (along the airway) occurs at an intermediate spacing of $D_c = 5D$. Interestingly, both transverse ($y$–direction) and vertical transport ($z$–direction) decrease monotonically with increasing inter-cluster spacing, with the maximum transverse and vertical transport observed when the clusters are next to each other.

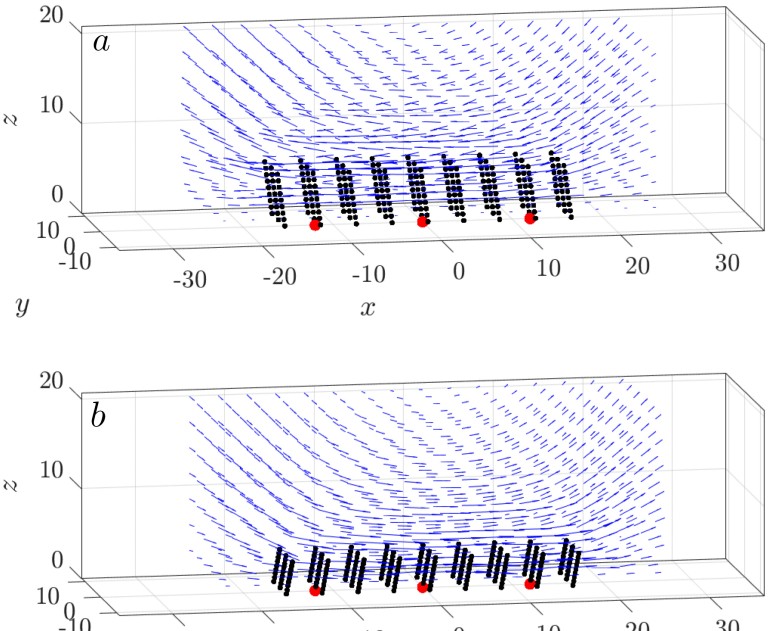

**Fig 14**. **Uniform horizontal motion in one large cluster form by placing three cilia clusters next to each other, $D_c = 3D$.** The 3-dimensional velocity field and the orientation of all cilia at $\phi = \pi/2$ (a), and $\phi = 3\pi/2$ (b). The red dots indicate the cluster centers. All cilia beat synchronously.

Fig 18 shows the vorticity field in the *xy*-plane for varying cilia cluster spacing, $D_c \in [3D,\ 12D]$. Each cilia cluster generates a localized zone of high vorticity centered around it, corresponding to the rotational flow. At the smallest spacing $D_c = 3D$, the three clusters effectively function as a single elongdated cluster, with their vorticity fields strongly overlapping. These high-vorticity zones are narrowly confined along the *x* direction, spanning approximately $[-20, 20]$ $\mu m$. As spacing increases to $D_c = 5D$ and $7D$, the clusters decouple spatially, but their vorticity fields remain partially connected, forming hydrodynamically coupled zones that allow for fluid and particle transfer between clusters. At larger spacings ($D_c = 9D$ and $12D$), the vorticity zones become fully separated and spatially isolated. This loss of coupling limits inter-cluster hydrodynamic interactions and reduces coordinated transport across the tissue surface, effectively confining rotational flow to small regions around each cluster.

Fig 19 shows Ripley's K function analysis for particles in three cilia clusters with varying cluster spacing from $D_c = 3D$ to $12D$. We focus on particles near the cilia tip at a height of $z = 7\mu$m. The results show evident short-range particle aggregation and long-range dispersion across all spacings. The sizes of these aggregates correlate with the swirl sizes and increase with increasing $D_c$, reaching a plateau for spacing greater than $D_c = 10D$.

Taken together, the swirls generated by individual cilia clusters interact hydrodynamically, enhancing both directional transport and mixing. Importantly, there is an optimal spacing between the cilia clusters that maximizes the speed of directional transport when all cilia beat synchronously.

## Metachrony reduces clearance and enhances mixing

Thus far we have focused on synchronous cilia beating. On the airway surface with many cilia, the actuating organelles can coordinate with each other and collectively beat in the form of metachronal waves, where neighboring cilia beat sequentially (i.e., with a phase lag) rather than synchronously [44]. Metachronal waves in mammals can move 4–8

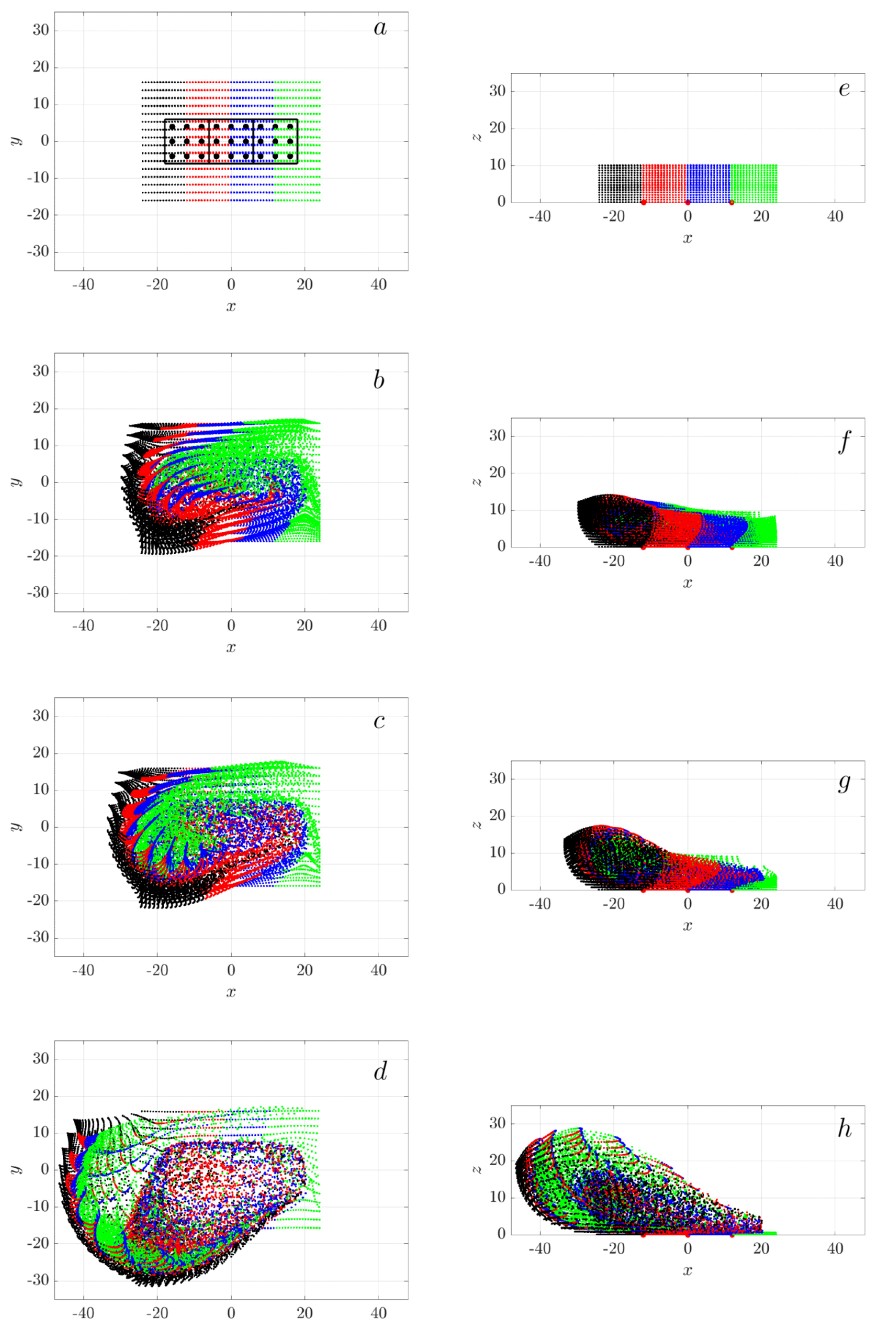

**Fig 15. Transport and mixing of particles in a cluster of three adjacent cilia clusters, separated by $D_c = 3D$.** The left column (a-d) shows a top view, and the right column (e-h) shows a side view. From top to bottom, sequential snapshots represent time points: $t = 0s$ (a,e), $t = 1.38s$ (b,f), $t = 2.78s$ (c,g), and $t = 11.11$ s (d,h), corresponding to 0, 50, 100, and 400 cilia revolutions, respectively. All cilia movesynchronously. Each cluster is marked by black boxes. Particle colors indicate their initial positions along the *x*-axis, illustrating the extend of horizontal transport and mixing.

times faster than mucociliary transport. In the trachea, these waves can move at speeds of 6–20 mm per minute [16]. The cilia metachrony is characterized based on the phase lag between each other and it arises via hydrodynamics

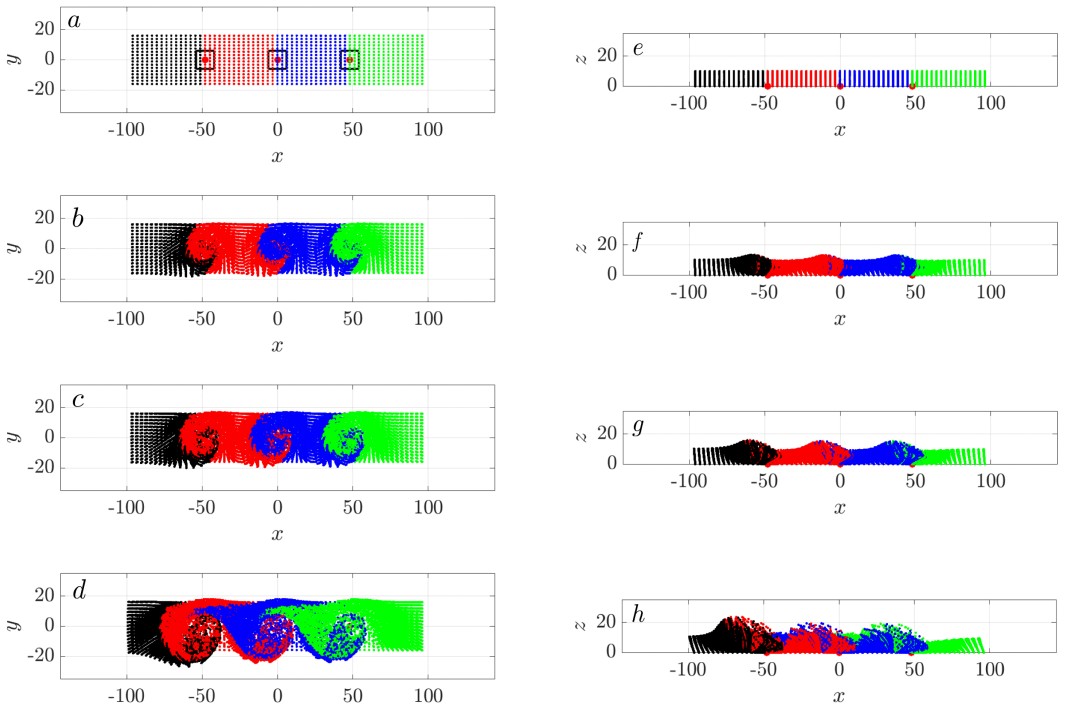

**Fig 16**. **Particle swirls from three widely separated cilia clusters, spaced at $D_c = 12D$.** The left column shows the top view (a-d) and the right column shows the side view (e)-(h). From top to bottom, sequential snapshots represent time points: $t = 0s$ (a,e), $t = 1.38s$ (b,f), $t = 2.78s$ (c,g), and $t = 11.11$ s (d,h), corresponding to 0, 50, 100, and 400 cilia revolutions, respectively. All cilia beat synchronously. Cluster centers are marked with red dots. Particle colors represent their initial positions along the $x$-axis, illustrating the dynamics of mixing and transport.

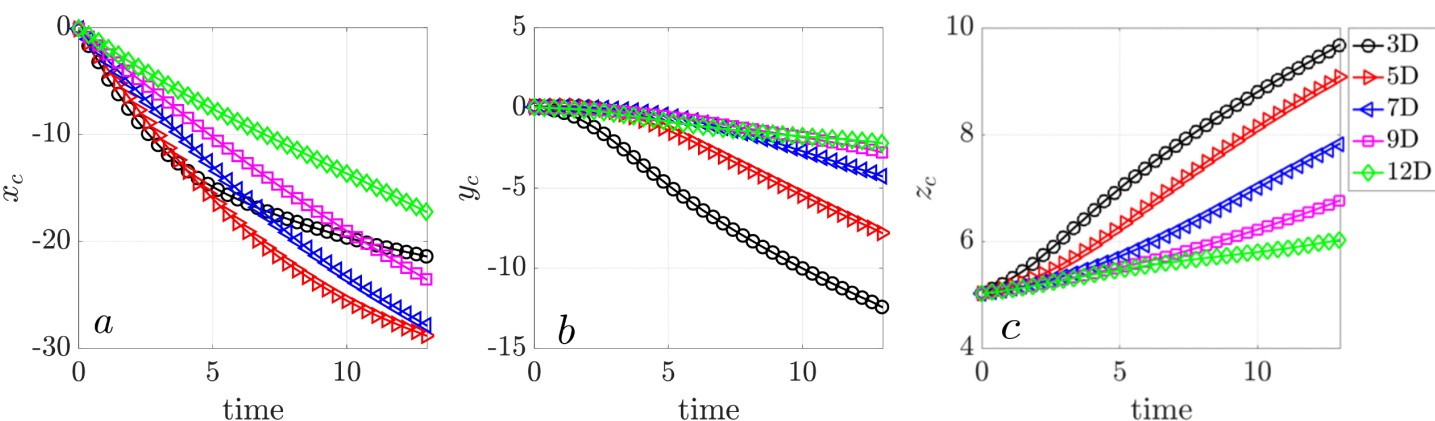

**Fig 17**. **Maximal transport at intermediate cluster spacing.** The trajectories of the center of mass (CoM) for all particles for three cilia clusters, in the (a) $x$-direction, (b) $y$-direction, and (c)$z$-direction. All cilia beat synchronously.

interactions [31]: symplectic waves move in the same direction of the effective stroke ($\phi_0 \in [-\pi, 0]$), while antipleptic waves move in the opposite direction (phase lag $\phi_0 \in [0, \pi]$).

Here, we assume that the metachronal wave is generated by the actuation of organelles rather than emerging from hydrodynamic interactions between beating cilia. We study the effect of metachronal wave on mucociliary transport by

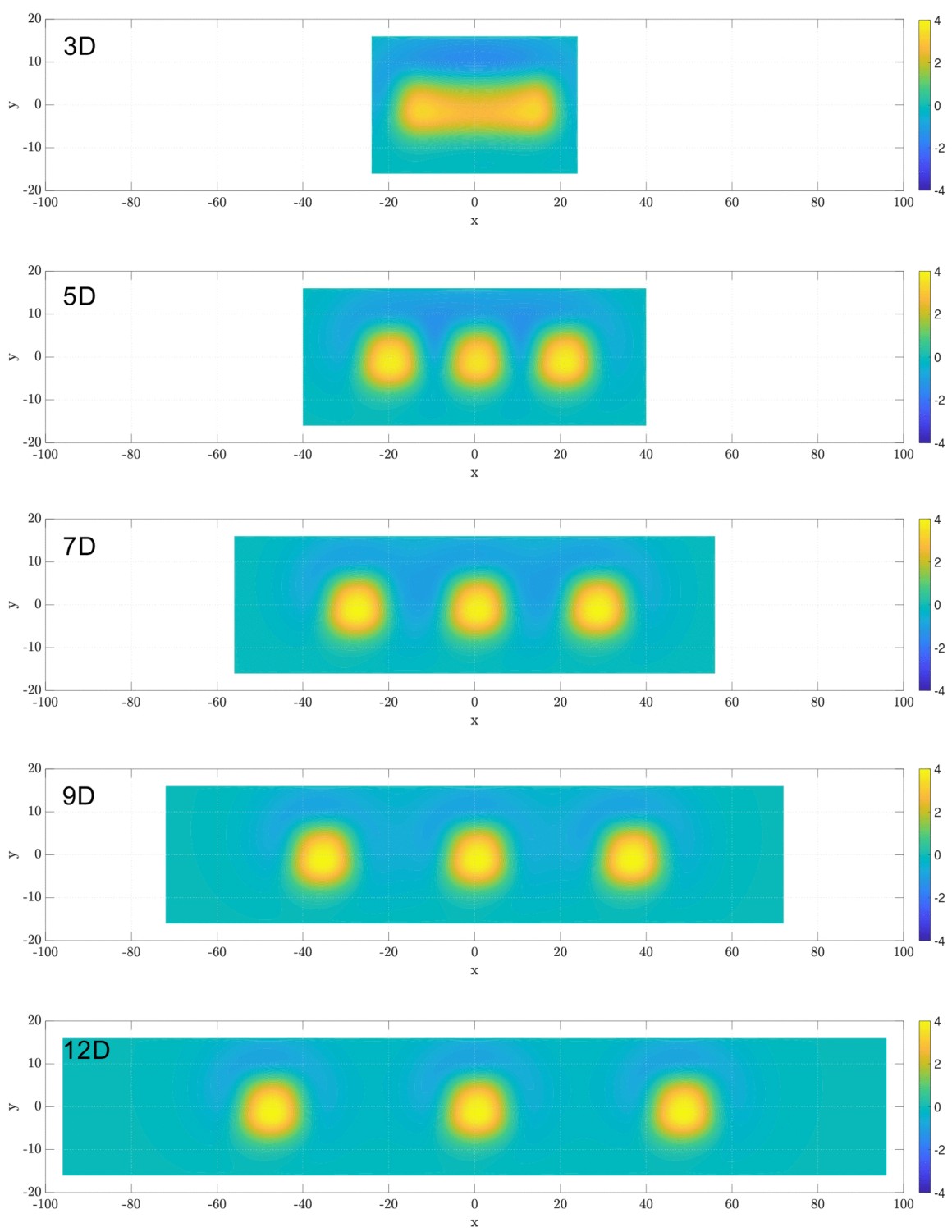

**Fig 18. Top-view vorticity fields at varying cilia cluster spacings $D_c \in [3D, 5D, 7D, 9D, 12D]$.** Each cilia cluster generates a localized zone of high voticity, corresponding to rotational flow. At the small spacing ($D_c = 3D$), the clusters connect to a single elongated cluster and their vortex zones merge into a single extended zone confined primarily along the $x$-axis. As spacing increases to $5D$ and $7D$, the vorticity zones begin to separate but remain partially connected, forming hydrodynamically coupled flow zones. At the largest spacings ($9D$ and $12D$), the high-vorticity regions are fully isolated, reducing inter-cluster interactions and coordinated transport.

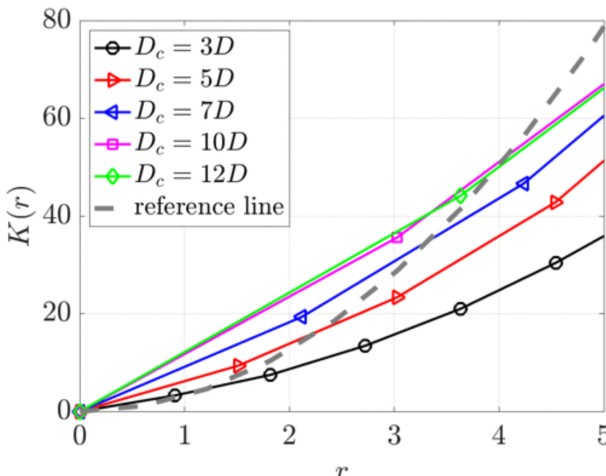

**Fig 19.** **Ripley's K function analysis of particle distributions in three cilia clusters with varying spacing from $D_c = 3D$ to $12D$.** This analysis focused on particle positions at a height of $z = 7\,\mu m$. The gray dashed reference line indicates complete spatial randomness.

imposing the phase lags directly between cilia. We focus on the scenario of three clusters with a cluster spacing of $D_c = 3D$, varying the phase lag $\phi_0$ between cilia. Results for other values of $D_c$ exhibit similar trends and are therefore not shown. The phase lag is applied to the cilia arrays in the $x$-direction. Specifically, each $3 \times 3$ cilia cluster is organized into 3 columns based on their $x$ locations. Each column begins with an equal increment of $\phi_0$, while maintaining synchronicity among the cilia within the same column.

With metachronal waves, the top-down view of particles at $t = 11.11$ s, corresponding to 400 cilia cycles, show reduced net transport and increased mixing (Fig 20). For $\phi_0 \neq 0$, particles aggregate into a large swirl. As the phase lag increases, more particles become confined to the central region, where they are well-mixed, and the net transport of particles in the negative $x$ and $y$ directions progressively diminishes. At $\phi_0 = 2\pi/3$, all particles are effectively trapped, and the swirl transforms into an ellipse. This occurs because the phase lag is commensurate with the cilia array's periodicity: after 3 columns, the phase lag totals $2\pi$, equivalent to no phase lag between clusters. Thus, the metachronal wave suppresses directional transport in the $x$–$y$ plane within a large cilia cluster, similar to the big swirls observed in Loiseau et al. [39].

We examine how metachronal coordination influences fluid transport in a system of three cilia clusters, focusing on varying spacing $D_c = 3D, 5D, 7D$. Since the physiological clearance direction corresponds to the negative $x$-direction, transport along this direction is of primary interest.

We first analyze the center-of-mass (CoM) trajectories of particles in the $x$, $y$, and $z$ directions for tightly packed configuration ($D_c = 3D$), for varying phase lags $\phi_0 \in [-2\pi/3, 2\pi/3]$ (Fig 21a). The black curve marks synchronous beating ($\phi_0 = 0$). In the antiplectic regime ($\phi_0 > 0$), transport along the negative $x$-axis gradually weakens with increasing phase-lag and ultimately reverses direction at $\phi_0 = \pi/3$, opposing clearance. The transport nearly vanishes at $\phi_0 = 2\pi/3$, indicating minimal net transport. In contrast, symplectic metachrony ($\phi_0 < 0$) shows modest enhancement in clearance for small phase lags ($\phi_0 = -\pi/12$ and $\pi/5$), with larger phase-lags showing performance similar to synchrony.

Lateral ($y$-axis) transport ($y_c$) shows a distinct trend: antiplectic waves enhance transport in the negative $y$-direction, while symplectic waves either reduce it or reverse the direction, depending on the lag. Small phase-lag ($\phi_0 = -\pi/12$) suppresses negative $y$ displacement, while larger symplectic lags lead to reversal. A phase lag of $\phi_0 = 2\pi/3$ results in minimal change.

Vertical transport ($z$-axis) behaves differently. While $z_c$ increases over time for all cases, both antiplectic and symplectic coordination reduce vertical displacement relative to synchrony. Antiplectic waves more strongly suppress vertical motion,

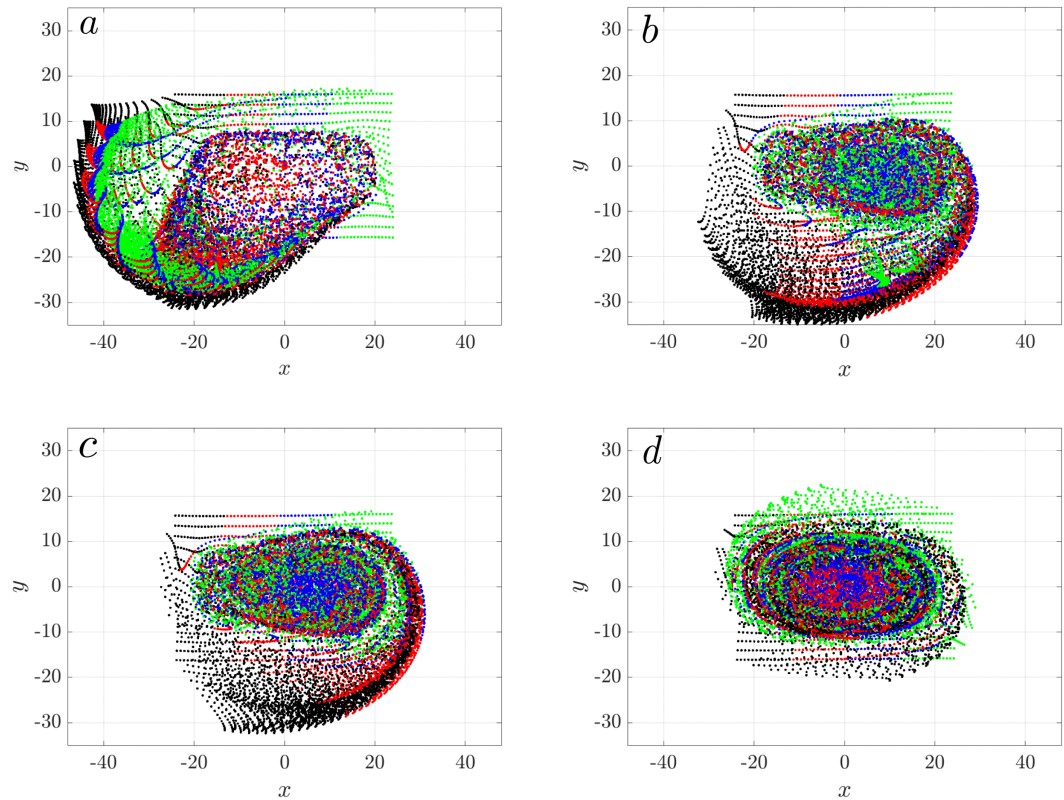

**Fig 20. Metachronal wave reduces directional transport in three connected cilia clusters with spacing $D_c = 3D$.** Snapshot of massless particles at $t = 11.11$ s are shown for varying phase shifts applied along the $x$ direction. (a) $\phi_0 = 0$, (b) $\phi_0 = \pi/4$, (c) $\phi_0 = \pi/3$, (d) $\phi_0 = 2\pi/3$. At $\phi_0 = 2\pi/3$, the three columns exhibit a metachronal wave with a wavelength of $2\pi$. Particles are color-coded based on their initial positions in the $x$ direction at $t = 0$, same as in Fig 15a.

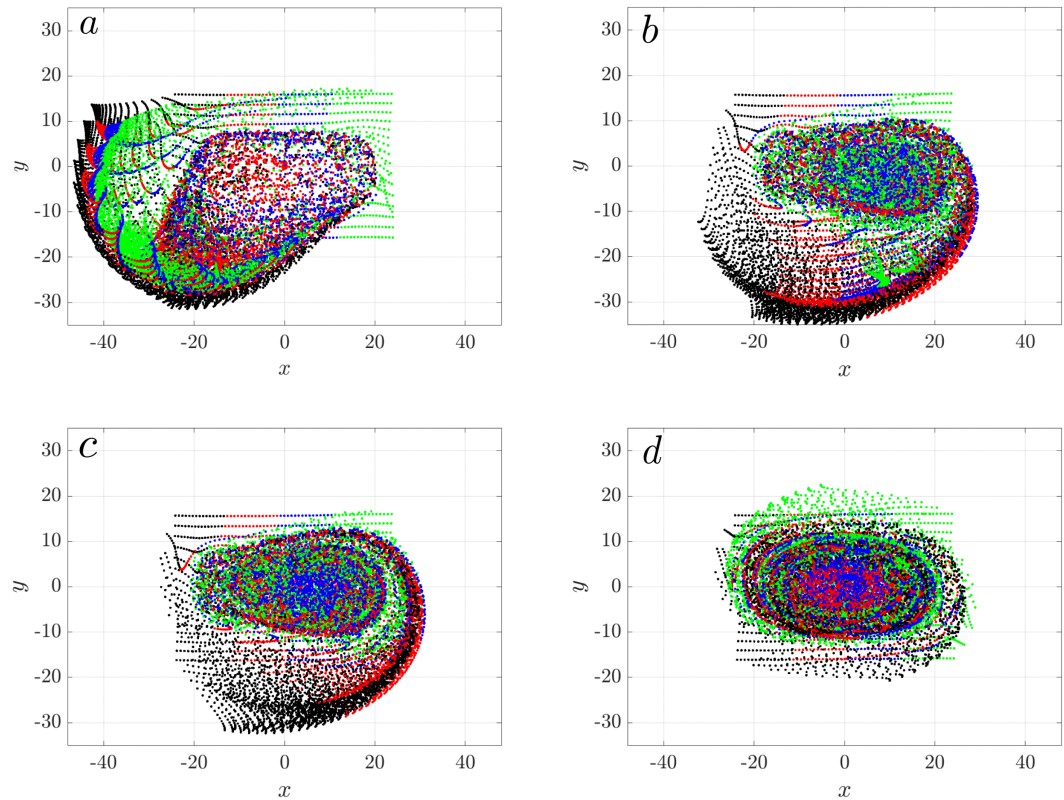

suggesting that synchrony drives the most effective upward transport near the cilia tips. This is consistent with the emergence of vertical vortices induced by metachronal waves, which redirect flow in more complex 3D patterns as seen in Ding et al. [7].

Interestingly, for $\phi_0 = \pm 2\pi/3$, particles show minimal net displacement in the $x$-$y$ plane, yet continue to move rapidly up in the $z$ direction. As shown in Fig 20d, these particles remain confined within a large coherent swirl, effectively trapped laterally while being lifted vertically. Because of the 3-row cilia cluster design, $\pm 2\pi/3$ corresponds to a rotating vortex flow that drive particles upward.

We next examine how changing the inter-cluster spacing influences the impact of metachronal wave on clearance. We focus on CoM in the $x$ direction. At $D_c = 5D$ (Fig 21b), the optimal spacing for synchronized cilia beating, none of the tested metachronal waves improve transport efficiency. In fact, antiplectic waves with $\phi_0 = \pi/3$ and $2\pi/3$ generate reversed flow, impeding clearance. For larger spacing ($D_c = 7D$, Fig 21c), again metachronal waves prove less effective than synchrony. The ability of antiplectic waves to reverse flow weakens with increasing spacing, and no metachronal configuration outperforms synchrony in driving clearance.

As metachronal coordination induces vertical displacement of particles, we expect it to increase vertical mixing. Figs 22a-d illustrates this effect for a system of three clusters spaced at $D_c = 3D$, with a representative antiplectic phase lag $\phi_0 = \pi/3$. Initially, particles are divided into two distinct vertical groups (black and cyan). Over time they become increasingly mixed. Some particles remain segregated after 400 beating cycles.

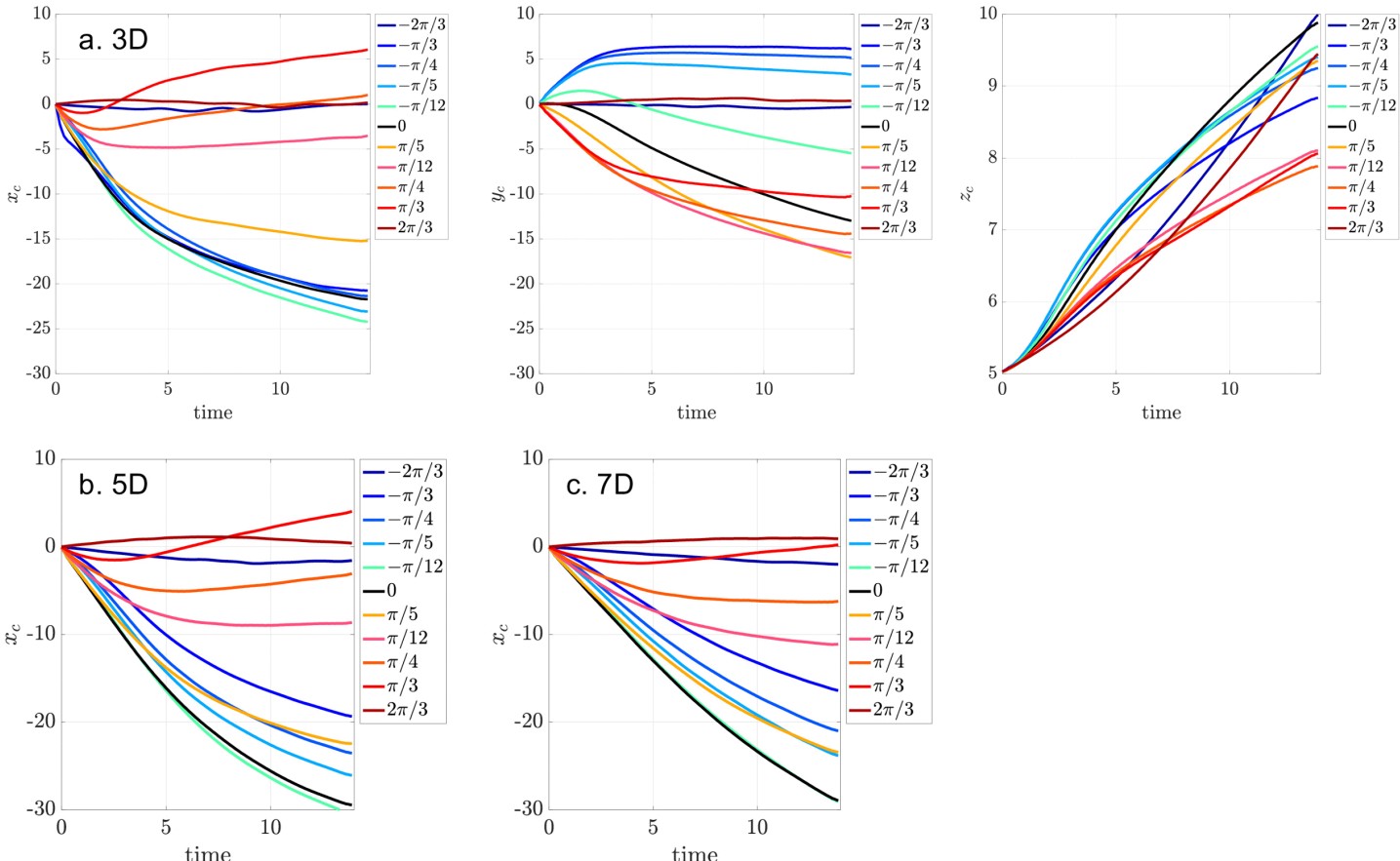

**Fig 21.** **Center-of-mass displacement for massless particles in a system of three cilia cluster with metachronal coordination, shown for phase lags $\phi_0 = 0, \pm\pi/12, \pm\pi/5, \pm\pi/4, \pm\pi/3, \pm2\pi/3$, under three inter-cluster spacings: (a) $D_c = 3D$ ($x_c, y_c, z_c$), (b) $D_C = 5D$ ($x_c$ only), and (c) $D_C = 7D$ ($x_c$ only).** The black curve corresponds to synchronous beating ($\phi_0 = 0$).

To quantify this process, Fig 22e shows using the normalized mixing number as a function of time for various phase lags. All curves show a monotonic decrease, consistent with progressive mixing. The rate of decay slows over time as the system approaches a well-mixed state. The black curve represents synchronous cilia beating. While all forms of metachrony accelerate mixing, antipletic waves ($\phi_0 > 0$) produce faster mixing than sympletic waves ($\phi_0 < 0$).

While simulating the full 400-beating-cycle dynamics at the tissue scale for three ciliary densities with all metachrony cases is computationally prohibitive, we present the time-averaged velocity fields near the cilia tip over one cycle to illustrate the flow coordination (Fig 23). We vary cilia densities ($\nu = 0.1, 0.2$, and $0.4$) and phase lags ($\phi_0 = -2\pi/3, -\pi/3, \pi/3$, and $2\pi/3$). Compared to the synchronous case (Figs 10d-f), the metachronal configurations produce qualitatively similar vortex structures, characterized by strong circular motions near each cilia cluster (centers marked by red dots) and increased velocity magnitude at higher cilia density ($\nu$). The key difference lies in the direction of the velocity vectors: metachronal waves substantially alter the transport direction. In particular, the velocities in negative $x$-direction are smaller than in synchronous beating, especially with $\pi/3$, the flow direction reverses.

Our results thus far demonstrate that metachronal coordination, while known to facilitate global flow organization, often reduces clearance transport in small cilia cluster systems, in some cases even reverses the flow direction. At the same

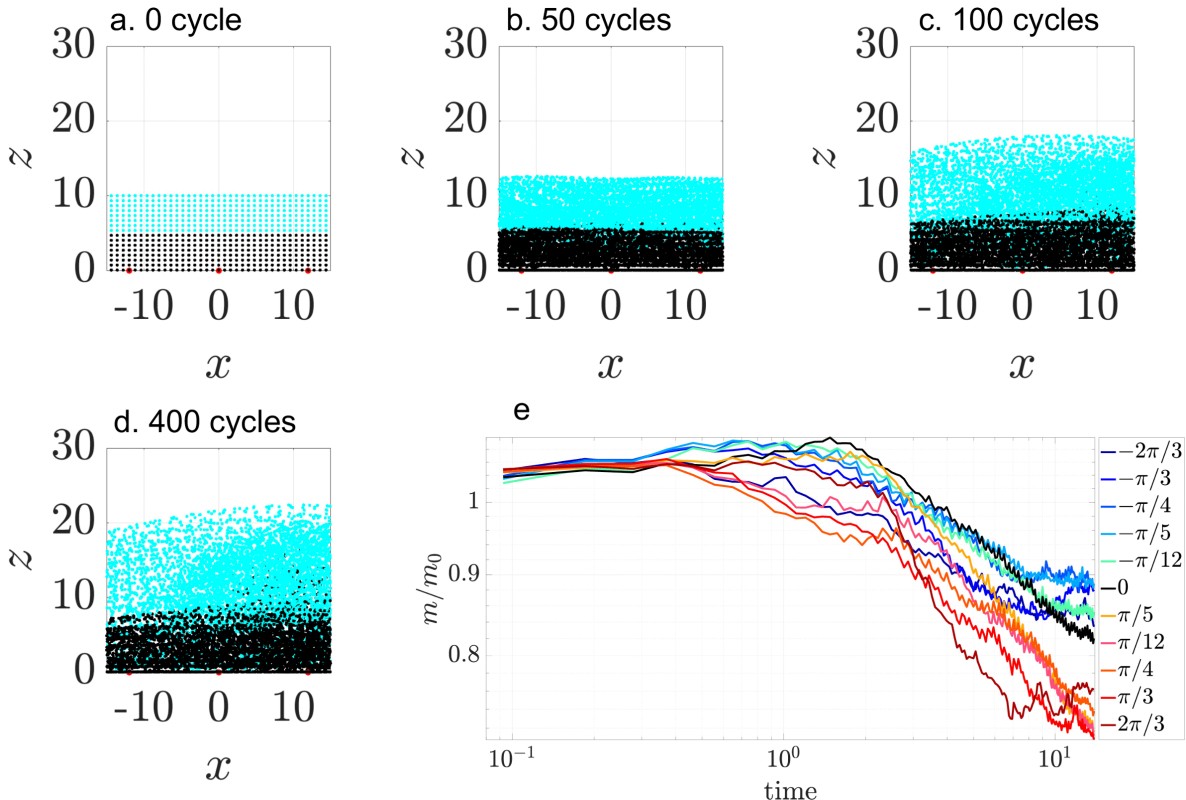

**Fig 22.** **Vertical mixing of passive particles over time in a three-cluster cilia array.** Particles are initially divided into two vertical groups: the bottom half (black) and the top half (cyan) to visualize vertical stratification and subsequent mixing. (a-d) Snapshots of particles after 0,50,100,200, and 400 cycles of cilia beating, shown for a phase lag of $\phi_0 = \pi/3$. (e) Temporal evolution of the normalized mixing number $m/m_0$ on a log-log scale for varying phase lag $\phi_0 \in [-2\pi/3, 2\pi/3]$. Lower values of $m/m_0$ indicate greater vertical mixing. The black curve represents synchronous beating. Colored curves correspond to metachronal waves with varying phase lags.

time, it consistently accelerates vertical particle displacement, enhancing vertical mixing. This trend holds across different inter-cluster spacings, though the magnitude of the effect reduces with increased spacing.

## Discussion

The ciliated cells of the airway epithelium play a crucial role in facilitating mucociliary clearance, which helps clear the airway of pathogens and other particles. In this study, we use computational tools to simulate cilia activity and transport of a viscous fluid on a small tissue scale, and examine the transport and mixing of tiny particles on a tissue patch, an array of three ciliary clusters, and then a single ciliary cluster.

The size of a tissue patch is comparable to that of human bronchial culture in a recent experimental study [38]. In this case, beating cilia induce swirling flow patterns, similar to those observed in human bronchial tissue experiments [38,41]. We found that transport is indeed enhanced at higher ciliary densities. However, interestingly, the size of particle clustering or swirling patterns was found to be independent of the ciliated density. This contrasts with findings from Gsell et al. (2020) [30]. The circular pattern induced by a patch of cilia clusters are also observed in the experimental study of Sears et al. (2015) [48] using primary human airway epithelial cells and referred to as mucus hurricane. Such observations have been instrumental in investigating the pathogenesis of cystic fibrosis [11]. These swirling patterns and their implications in mucus transport dynamics highlight the intricate interplay between ciliary motion, fluid dynamics, and respiratory health.

**Fig 23. Time-averaged velocity field near the cilia tip ($z = 7\,\mu m$) at the tissue scale under metachronal coordination.** Each panel corresponds to a different ciliary density $\nu = 0.1, 0.2$, and $0.4$, with an associated phase lag ($\phi_0 = -2\pi/3, -\pi/3, \pi/3$, and $2\pi/3$). The velocity fields are averaged over one full cilia beating cycle. Red dots mark the centers of cilia clusters.

While many studies have investigated the formation of metachronal waves and their role in either facilitating or hindering the efficiency of fluid pumping [5,19,24,26,43,53], the results remains inconclusive. For example, Gauger et al. [26] modeled cilia with super-paramagnetic particles in an external magnetic field and found that antipleptic metachrony was more efficient for transport with a particular cilia spacing. Wollin and Stark [53] modeled cilia as chains of beads and found transient synchronization in a bulk fluid. Elgeti and Gompper [24] represented cilia as semi-flexible rods with active bending forces, identified both symplectic and laeoplectic (perpendicular to the power-stroke direction) metachrony, and noted a 10-fold increase in propulsion efficiency compared to synchronous beating.

Our model uses prescribed ciliar beating and metachronal coordination. Interestingly, we observe that most metachronal coordinations reduce clearance transport, with some antiplectic waves inducing reverse transport in the negative $x$-direction, as measured by the center of mass of the tracer particles. This finding contrasts with several previous reports where metachronal waves generally enhanced transport efficiency (e.g., [5,7,19], except for a few antiplectic cases where flow reverses direction [19].

Three factors may explain this discrepancy. 1) The local metachronal coordination: In our model, the metachronal wave propagates along the $x$–direction within each cilia cluster, with a fixed phase-lag $\phi_0$ between adjacent cilia columns. Our cilia only occupy a truncated portion 8 $\mu m$ of the wavelength. Between cilia clusters, especially at low ciliary densities, the inter-cluster spacing is more likely incompatible with the metachronal wavelength, leading to destructive interference.

2) Open boundary condition: Unlike most previous models that use doubly periodic boundary conditions, our model has open boundary conditions in $x,y$ and $+z$ directions, allowing tracer particles to move outside the simulation box. This setup makes the volume-based velocity averaging impossible.

3) Transport quantification: Previous studies typically measure transport using global displaced fluid volume per cilia beating cycle, based on the volume-averaged flow rates. We track the tracer particle trajectories directly. The differences may limit the direct comparison of results.

Understanding the characteristics of particle transport and mixing within the mucociliary system is crucial for comprehending disease pathophysiology and developing effective treatment strategies. By precisely prescribing the motion of 3D cilia, we can isolate and study the effects of various parameters, thereby taking a fundamental step toward understanding the transmission of particles such as bacteria, viruses, and inhaled drugs.

Our idealized constitutive model of mucus and cilia motion provides a computational platform for examining the fluid dynamic consequences of ciliary density, spacing, and metachrony in simplified fluid environments. However, we emphasize that our simulations are not designed to replicate physiologically accurate mucociliary clearance in the human airway. First, mucus is a viscoelastic gel, its polymeric network property deviates substantially from the Newtonian fluid assumption used here. Second, we adopt a simplified representation of cilia as rotating rigid rods, which does not capture the asymmetric power and recovery strokes observed in vivo. These limitations mean that the modeled flows likely differ qualitatively from actual mucus transport. Nevertheless, the insights obtained from this idealized system may still inform the development of artificial cilia arrays and inspire hypotheses for future models that include non-Newtonian fluid properties and fully resolved ciliary kinematics. Lastly, the ciliary patch placement in airway tissue is intrinsically stochastic. While we used three independent simulations of cilia cluster placements, additional realizations may strengthen the statistical robustness of our findings.

## Data availability

The source code and simulation data used to produce the simulations and analyses are available in the GitHub repository: https://github.com/Jiang-Lab/Mucociliary_Clearance-Tissue_Scale_model.

## Acknowledgments

YJ acknowledges the support from the Frady Whipple Professorship. We thank the Advanced Research Computing Technology & Innovation Core (ARCTIC) team at Georgia State University for computational support. Ling Xu would like to thank the University of North Carolina at Chapel Hill and the Research Computing group for providing computational resources and support that have contributed to these research results.

## Author contributions

**Conceptualization:** Ling Xu, Yi Jiang.

**Data curation:** Ling Xu, Yi Jiang.

**Formal analysis:** Ling Xu, Pejman Sanaei, Yi Jiang.

**Funding acquisition:** Yi Jiang.

**Investigation:** Ling Xu, Pejman Sanaei, Yi Jiang.

**Methodology:** Ling Xu, Yi Jiang.

**Project administration:** Yi Jiang.

**Resources:** Ling Xu, Yi Jiang.

**Software:** Ling Xu.

**Supervision:** Ling Xu, Yi Jiang.

**Validation:** Ling Xu, Yi Jiang.

**Visualization:** Ling Xu, Pejman Sanaei, Yi Jiang.

**Writing – original draft:** Ling Xu, Yi Jiang.

**Writing – review & editing:** Ling Xu, Pejman Sanaei, Yi Jiang.

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
