## [Decision Letter · Decision Letter 0]

12 Mar 2025

PCOMPBIOL-D-24-02256

Modeling mucociliary mixing and transport at tissue scale

PLOS Computational Biology

Dear Dr. Jiang,

Thank you for submitting your manuscript to PLOS Computational Biology. After careful consideration, we feel that it has merit but does not fully meet PLOS Computational Biology's publication criteria as it currently stands. Therefore, we invite you to submit a revised version of the manuscript that addresses the points raised during the review process.

In particular, we would request to pay particular attention to the reviewer comments about the biological signficance and generating predictions (going beyond observations). 

Please submit your revised manuscript within 60 days May 11 2025 11:59PM. If you will need more time than this to complete your revisions, please reply to this message or contact the journal office at ploscompbiol@plos.org. Please include the following items when submitting your revised manuscript:

We look forward to receiving your revised manuscript.

Kind regards,

Douglas Brumley

Academic Editor

PLOS Computational Biology

Padmini Rangamani

Section Editor

PLOS Computational Biology

**Additional Editor Comments :**

We have now received the reports from three Reviewers for this manuscript. There are a range of significant points raised, in particular related to the qualitative nature of many of the results; technical foundations of the analysis and mixing efficiency, and mechanistic understanding of the dynamics with metachronal waves. There were also several points raised about the advancement of new knowledge beyond previously published work.

I invite the authors to submit a "major revision" to their paper, carefully addressing all of the issues raised by three Reviewers. Please include a point by point response to every comment, and highlight in a revised paper where any changes have been made. Given the substantial nature of the points raised, there is no guarantee that this would lead to acceptance. Please don't hesitate to get in touch if you have any questions.

**Journal Requirements:**

At this stage, the following Authors/Authors require contributions: Ling Xu, Pejman Sanaei, and Yi Jiang. Please ensure that the full contributions of each author are acknowledged in the "Add/Edit/Remove Authors" section of our submission form.

3) Your manuscript is missing the following section: Methods.  Please ensure all required sections are present and in the correct order. Make sure section heading levels are clearly indicated in the manuscript text, and limit sub-sections to 3 heading levels. An outline of the required sections can be consulted in our submission guidelines here:

**Reviewers' comments:**

Reviewer's Responses to Questions

**Comments to the Authors:**

**Please note that one of the reviews is uploaded as an attachment.**

Reviewer #1: Strengths

+ The motivation is clear and the paper is well-written. Experimental results on ciliary sheets are an emerging area, with intrinsic interest due to the relevance of pumping in mucus clearance.

+ The technical work is high-quality. I agree with the authors’ computational approach. IB with rigid rotors is a good approximation for this system, and I like the trick of modifying the effective length, in order to account for the asymmetry of the forward and return stroke. I do not think that a finer level of modeling detail (fluid-structure, force feedback) is necessary for this problem scale.

+ The authors pick most of their modeling parameters (like spacing between cilia) based on published values, and they vary others over biologically-relevant scales, such as the distances between cilia clusters.

Weaknesses

While I like the problem choice and approach, I’m not as excited about the results, which could be richer and thus more informative for biological systems.

+ The description of the mixing results is overall qualitative, in terms of the number of “swirls” and their relative placement. There are several schematics of flow fields, which are informative, but which would be clearer if they demonstrated an overall scaling trend. I would be interested in a more holistic quantitative measure of mixing quality. The MSD doesn’t appear to be sensitive to the qualitative changes in mixing effects that the authors report. The complex 3D mixing dynamics in the full simulations shown in Fig. 5 is a phenomenon worth exploring more deeply.

+ I would prefer a more hydrodynamic analysis of the simulation results. From a hydrodynamics perspective, the ciliary carpet injects vorticity into the flow at the patch tips. These vorticity patches appear to combine across scales, giving rise to larger-scale vorticity patches away from the cilia tips. Can we say something more detailed about the relevant scales for mixing and clearance for this system?

+ I’m a bit concerned that the “swirls” are a transient effect, particularly because they appear to coarsen over time. Does the flow reach a statistical steady-state, where fluctuations in vorticity, scalar dispersion, etc have a well-defined average? This would help differentiate the effects of spacing/placement conditions.

+ For the problems of pumping under synchrony and metachronal waves, the simulation setting seems simplified compared to the more elaborate random X-Y placement of patches in the earlier results (Fig 5). Enhanced transport due to high cilia density is not surprising in itself, since additional cilia inject more power into the flow. I’d be more curious how we might understand this phenomenon in a more complex geometry, particularly in light of the complex eddies that the authors observe when the cilia patches are placed randomly?

Other questions:

+ I'm confused regarding the finding that metachronal waves inhibit transport. Doesn't this depend very sensitively on the beating rate, phase offsets, spacing, flow rate, etc?

+ Are all cilia in all clusters always aligned (pumping the same direction)?

+ Why is enhanced mixing or particle dispersion desirable in this system? In a biological system such as the lungs, is net transport the only readout that matters?

+ The authors observe a bias towards counter-clockwise rotation, and the net flow has a slight diagonal drift. What is causing this effect? In my reading, I didn't see what breaks the symmetry in the system.

+ Are cilia clusters known to be distributed randomly? Or do they typically appear in evenly-spaced rows, like those considered in the metachronal wave and synchronization results?

Reviewer #2: This paper is extremely well written, providing a significant step forward beyond results available in the literature on an important problem in respiratory physiology: mucociliary clearance of the barrier fluid coating the respiratory tract.

The paper summarizes "target data and information" with respect to existing experimental and clinical data on cilia propulsion of mucus, and uses that data as a basis for their modeling approach. This is "all good" and represents a nice contribution to the literature.

However, there are two essential missing ingredients in their model of mucociliary clearance.

One: The authors make the assumption that has been made by the preponderance of the historical literature on modeling of cilia propulsion of mucus. This history includes cilia forcing of the mucosal barrier at the single cilium scale as well as attempts to explain synchronization of cilia and propulsion at the cellular (~200 cilia) and tissue scales. Namely, the foundation of this paper and most previous modeling of cilia propulsion of mucus is based on the assumption that the mucus layer is an incompressible viscous fluid. Mucus is not a viscous fluid; it is an entangled polymer solution whose properties are not remotely approximated by the Stokes equations as done here (and others in previously published work).

Two: the cilia beat stroke consists of a power stroke in which the cilium is approximately rigid and rod-like and penetrates the mucus layer by one micron at the peak of the power stroke, but the return stroke is quite curved and does not make contact with the mucus layer. That stroke is far from the model in this paper, and the flows generated will almost surely be wildly different even in a viscous fluid. So the beautiful modeling done in this paper does not shed light on the transport of mucus in the annular "fluid layer" between the cellular tissue and the air core.

That said, to their credit, there are several important details present in the paper that are extensions of what has been done previously. So I am happy to support publication if the title and the claims are walked back to something along the lines of:

"modeling of ciliary transport of viscous fluids in airway-like geometries". Then one can say what is new here, and that the results need to be extended significantly to "solve" the real problem of physiologically relevant mucociliary clearance. Otherwise readers might think this paper is relevant to the human respiratory tract -- and with all due respect, it is not! I can say, with absolute certainty, that the flows generated by beating cilia at the cellular and tissue scale are wildly different for viscous fluids and mucus. It is misleading to suggest otherwise.

Furthermore,the diffusion of small molecule concentrations performed in this paper will not be relevant to airway surface liquids and drug delivery .... the advection-diffusion will be strongly dominated by the correct 3D advection velocity field, on the upside, small molecule diffusion is not dramatically different since sufficiently small molecules will diffuse normally and not undergo diffusion with memory as large molecules or particles in complex polymeric fluids.

If the authors state very clearly that this idealized constitutive model of mucus in their modeling platform provides a foundation for extension to complex fluid models of mucus, and an idealized model of the cilium beat stroke, then the reader will understand that the relevance to the respiratory tract is unknown and this paper is giving a preview of what it might actually look like. NOBODY HAS DONE THIS IN 3D SIMULATIONS, even with viscous fluids, SO THE RESULTS IN THIS PAPER ARE AN IMPORTANT CONTRIBUTION! To become more relevant, one will need to derive a good entangled polymer model for mucus, and couple that constitutive equation to the Navier-Stokes or Stokes model as is done in all viscoelastic fluid solvers. This advance -- a good continuum scale constitutive equation of respiratory mucus, remains an open problem. There is a significant literature on microrheology of mucus that produces the equilibrium viscoelastic moduli of mucus, but it is still a large step to take that data and embed it into a macroscopic constitutive equation. That is itself an open problem. Another open problem is to use dynamic microscopy to get the full cilium beat stroke shape.

For this reason, the authors cannot be criticized for not modeling physiological mucociliary clearance.....that problem is not going to be solved anytime soon.

I recommend publication of this paper given the above limitations because this 3D computational platform is an advance and the platform will apply to mucociliary clearance simulations when generalized to the correct governing equations for mucus and realistic beat strokes of cilia, as stated above.

Reviewer #3: attachment

**Have the authors made all data and (if applicable) computational code underlying the findings in their manuscript fully available?**

Reviewer #1: **No: **The authors provide a GitHub link, but the code does not have documentation.

Reviewer #2: Yes

Reviewer #3: None

PLOS authors have the option to publish the peer review history of their article (what does this mean?). If published, this will include your full peer review and any attached files.

Reviewer #1: No

Reviewer #2: **Yes: **M. Gregory Forest

Reviewer #3: No

**Figure resubmission:**
---

## [Decision Letter · Decision Letter 1]

4 Aug 2025

PCOMPBIOL-D-24-02256R1

Modeling tissue-scale ciliary transport and mixing in three-dimensional Newtonian flow

PLOS Computational Biology

Dear Dr. Jiang,

Thank you for submitting your manuscript to PLOS Computational Biology. After careful consideration, we feel that it has merit but does not fully meet PLOS Computational Biology's publication criteria as it currently stands. Therefore, we invite you to submit a revised version of the manuscript that addresses the points raised during the review process.

Please submit your revised manuscript within 60 days Oct 04 2025 11:59PM. If you will need more time than this to complete your revisions, please reply to this message or contact the journal office at ploscompbiol@plos.org. Please include the following items when submitting your revised manuscript:

We look forward to receiving your revised manuscript.

Kind regards,

Douglas Brumley

Academic Editor

PLOS Computational Biology

Feilim Mac Gabhann

Editor-in-Chief

PLOS Computational Biology

**Additional Editor Comments (if provided):**

The authors have made substantial improvements to their manuscript in the revision process. However, there are still significant concerns about the stochastic nature of the simulations, and the fact that many conclusions are drawn from a small number (or single run) of simulations. I would encourage the authors to revise their manuscript addressing these and any other concerns. This could include conducting a series of additional simulations (average over more realisations), and showing the variance of the results across simulations with the same set of parameters.

**Journal Requirements:**

**Reviewers' comments:**

Reviewer's Responses to Questions

**Comments to the Authors:**

Reviewer #1: Thanks for the comprehensive revision, the new mixing metric and broader set of simulations are great. I am happy to recommend publication.

Reviewer #2: I have the same comments to the authors as I have to the Editor: The authors have convincingly incorporated all concerns and requests of the 3 Reviewers in the resubmission. This is a very nice advance of the previous literature on ciliary transport and mixing in 3D, and provides a foundation for the next major advance: to replace viscous incompressible fluids with physiologically relevant constitutive models of respiratory mucus.

Reviewer #3: I have read the revised manuscript "Modeling tissue-scale ciliary transport and mixing in three-dimensional Newtonian flow" and I have read the authors response to my comments.

I appreciate the authors' efforts to revise their work and cite the relevant work (e.g., "Mixing and transport by ciliary carpets: a numerical study"). Note that in their response letter, the authors mischaracterize the study of Ding et al to be 2D -- the fluid motion is 3D but the cilia beating is 2D. My major concerns about the study still hold. The study offers no new methods and no deep insights into the mechanisms that lead to the swirling motion in this set-up. Importantly, given the stochastic nature of the input, the study should employ rigorous Monte Carlo simulations and appropriate statistical metrics to draw meaningful conclusions, rather than relying on isolated, single-run observations. Add to this that the link to physiological function of such swirling fluid motion is weak, I am not very enthusiastic about this work. I leave the final decision to the editor.

**Have the authors made all data and (if applicable) computational code underlying the findings in their manuscript fully available?**

Reviewer #1: Yes

Reviewer #2: Yes

Reviewer #3: **No: **I did not check the data availability

PLOS authors have the option to publish the peer review history of their article (what does this mean?). If published, this will include your full peer review and any attached files.

Reviewer #1: No

Reviewer #2: **Yes: **M. Gregory Forest

Reviewer #3: No

**Figure resubmission:**
---

## [Editor Report · Decision Letter 2]

20 Oct 2025

PCOMPBIOL-D-24-02256R2

Modeling tissue-scale ciliary transport and mixing in three-dimensional Newtonian flow

PLOS Computational Biology

Dear Dr. Jiang,

Thank you for submitting your manuscript to PLOS Computational Biology. After careful consideration, we feel that it has merit but does not fully meet PLOS Computational Biology's publication criteria as it currently stands. Therefore, we invite you to submit a revised version of the manuscript that addresses the points raised during the review process.

Please submit your revised manuscript within 30 days Dec 20 2025 11:59PM. If you will need more time than this to complete your revisions, please reply to this message or contact the journal office at ploscompbiol@plos.org. Please include the following items when submitting your revised manuscript:

We look forward to receiving your revised manuscript.

Kind regards,

Douglas Brumley

Academic Editor

PLOS Computational Biology

Feilim Mac Gabhann

Editor-in-Chief

PLOS Computational Biology

**Additional Editor Comments (if provided):**

I have reviewed the revised manuscript and authors' responses, which have adequately addressed the concerns raised by the reviewers. It was noted by several reviewers that the source code and data could be viewed, and an earlier data sharing statement noted a GitHub site. However, the data provided must be deposited in an appropriate public repository with various requirements satisfied, e.g., stable identifier DOI, long-term data management plan, etc. Can the authors please review the data sharing policy for PLOS Computational Biology, including recommended repositories and repository criteria (see here: https://journals.plos.org/ploscompbiol/s/recommended-repositories) and update the data sharing statement to fully comply with the policy. We look forward to hearing from you again soon.

**Journal Requirements:**

**Reviewers' comments:**

**Figure resubmission:**
---

## [Editor Report · Decision Letter 3]

15 Dec 2025

Dear Dr. Jiang,

We are pleased to inform you that your manuscript 'Modeling tissue-scale ciliary transport and mixing in three-dimensional Newtonian flow' has been provisionally accepted for publication in PLOS Computational Biology.

Best regards,

Douglas Brumley

Academic Editor

PLOS Computational Biology

Feilim Mac Gabhann

Editor-in-Chief

PLOS Computational Biology

---

## [Editor Report · Acceptance letter]

PCOMPBIOL-D-24-02256R3

Modeling tissue-scale ciliary transport and mixing in three-dimensional Newtonian flow

Dear Dr Jiang,

I am pleased to inform you that your manuscript has been formally accepted for publication in PLOS Computational Biology. Your manuscript is now with our production department and you will be notified of the publication date in due course.

With kind regards,

Anita Estes
